# Co-Design Dedicated System for Efficient Object Tracking Using Swarm Intelligence-Oriented Search Strategies

**DOI:** 10.3390/s23135881

**Published:** 2023-06-25

**Authors:** Nadia Nedjah, Alexandre V. Cardoso, Yuri M. Tavares, Luiza de Macedo Mourelle, Brij Booshan Gupta, Varsha Arya

**Affiliations:** 1Department of Electronics Engineering and Telecommunications, State University of Rio de Janeiro, Rio de Janeiro 20.550-900, Brazil; 2Department of Systems Engineering and Computation, State University of Rio de Janeiro, Rio de Janeiro 20.550-900, Brazil; 3Department of Computer Science and Information Engineering, Asia University, Taichung 41354, Taiwan; 4Center for Advanced Information Technology, Kyung Hee University, Seoul 02447, Republic of Korea; 5Department of Electrical and Computer Engineering, Lebanese American University, Beirut 1102, Lebanon; 6Center for Interdisciplinary Research, University of Petroleum and Energy Studies, Dehradun 248007, India; 7Department of Business Administration, Asia University, Taichung 41354, Taiwan; 8University Center for Research & Development (UCRD), Chandigarh University, Chandigarh 140413, India; 9School of Computing, Skyline University College, Sharjah P.O. Box 1797, United Arab Emirates

**Keywords:** object tracking, template matching, swarm intelligence, image cross-correlation

## Abstract

The template matching technique is one of the most applied methods to find patterns in images, in which a reduced-size image, called a target, is searched within another image that represents the overall environment. In this work, template matching is used via a co-design system. A hardware coprocessor is designed for the computationally demanding step of template matching, which is the calculation of the normalized cross-correlation coefficient. This computation allows invariance in the global brightness changes in the images, but it is computationally more expensive when using images of larger dimensions, or even sets of images. Furthermore, we investigate the performance of six different swarm intelligence techniques aiming to accelerate the target search process. To evaluate the proposed design, the processing time, the number of iterations, and the success rate were compared. The results show that it is possible to obtain approaches capable of processing video images at 30 frames per second with an acceptable average success rate for detecting the tracked target. The search strategies based on PSO, ABC, FFA, and CS are able to meet the processing time of 30 frame/s, yielding average accuracy rates above 80% for the pipelined co-design implementation. However, FWA, EHO, and BFOA could not achieve the required timing restriction, and they achieved an acceptance rate around 60%. Among all the investigated search strategies, the PSO provides the best performance, yielding an average processing time of 16.22 ms coupled with a 95% success rate.

## 1. Introduction

Object tracking in videos has myriad range of applications across various domains. It is exploited in *(i)* surveillance and security, which helps detect suspicious activities, identify intruders, and provide evidence for investigations; *(ii)* autonomous vehicles, which helps make informed decisions for navigation, collision avoidance, and path planning [1]; *(iii)* augmented reality, which utilizes object tracking to overlay virtual objects or information onto the real-world environment [2,3]; *(iv)* human–computer interaction, in which object tracking enables gesture recognition, hand tracking, and body tracking; *(v)* robotics, where object tracking is used to perceive and interact with surrounding objects; *(iv)* and face detection, which involves identifying and locating human faces in images or videos. It is used in facial recognition, emotion analysis, biometrics, and video surveillance [4].

Image processing is an important tool to assist in decision making. The continuous monitoring of any environment, such as public areas and industrial parks, helps define better action strategies and decide the right moment to act, reducing risks and enhancing opportunities. The quality of the information resulting from image processing must be good enough to avoid errors in scenario evaluation during the planning of future actions and the definition of goals to be achieved. The time to obtain this information and process it is also at the basis for the success of any related action. In general, a slow search process always ends up delaying decision making, so that the information obtained may become obsolete or insufficient at the time of decision making. For instance, observing the airspace makes it possible to recognize the presence of aircraft and assess whether there are possible collision routes. Likewise, just as an airplane pilot observes the environment to plan a flight, a self-guided mobile artifact uses the images of the environment [5]. In a fire situation, the rapid pinpointing of the focus facilitates the activation of firefighting and rescues, reducing any negative impact [6]. In a shopping center, observing customer behavior can provide information regarding whether they are about to make purchases [7].

Image and video processing has provided advances in several important areas of research. The emergence of new sensors and new equipment capable of capturing, storing, editing, and transmitting images has accelerated the decision-making process, allowing the definition of strategies with lower risk and a higher success rate. The time to obtain true information and process it is directly responsible for the success of the action. Artificial intelligence can help speed up the execution of procedures, as a slow search process can delay decision making so that information may become obsolete or insufficient [8].

Image tracing is the process that involves searching, identifying, and tracking a previously chosen visual element in larger images [9]. The identification and estimation of the target’s position in successive images allows tracking its trajectory in the frames of a video as it moves through the environment. The first image in the sequence of frames requires a greater effort to locate, as there is not necessarily a tendency for the target to be positioned. For the following images, the location in the previous frame suggests the initial search position, in case the target remains stationary or the image acquisition system manages to keep it in similar framing throughout the video. This initial search condition allows restricting the target search within a neighborhood, which saves search time and tends to increase the chance of success.

A target, to be identified in the midst of a complex image, must have unique characteristics so that it can be distinguished from the background. In [9,10], features are analyzed for target identification, such as color, edges, and textures. Hence, it is possible to identify reference elements of the objects of interest or even separate parts of the image to seek similarity with the target. Object movement while tracking can cause changes in object appearance, such as variation in size. Therefore, mathematical transformations such as rotation and scaling must be applied to the target model in an attempt to offer more possibilities for comparison with the suspected objects. Regardless, the tracking process can be hampered by object occlusions, noise in the images, changes in ambient lighting, and deformations caused by complex object movements, making it difficult to verify image similarity [11]. The loss of information due to the 2D projection from a 3D scene also complicates the target identification process in images. The adopted model for the target must have spatial and appearance information. It constitutes the pattern to be located.

Template matching (TM) is one of the most applied methods to find patterns in images [12]. It basically consists of counting the occurrence of a smaller image, which represents the target, within another larger image, which represents the environment as a whole. Among the TM techniques, the verification of the normalized cross-correlation coefficient (NCC) is widely used due to its properties of invariance to global brightness changes in images [13]. However, it is computationally more expensive when using large images and/or a set of images. For this technique, the Pearson correlation coefficient (PCC), another name by which NCC is known, is computed repeatedly.

The computational processing of the search requires a complex and is hardware-resource-hungry. For dedicated hardware, the available physical resources, such as memory, processing capacity, and required energy, are limited. These limitations are far tighter when the hardware must deal with large volumes of data, as is the case for template matching in images. Therefore, it becomes necessary to design search systems that support the demand, based on the optimization of available resources. One possibility of image processing with optimized resource allocation, aiming to increase performance, is the use of a hardware/software co-design implementation, wherein only the computationally intensive components of the system are implemented in the hardware, while the remaining ones are implemented in the software.

The use of dedicated hardware helps the system perform the relevant PCC calculations more quickly. However, the proper selection of which sector of the image will be compared with the target is essential for fast localization. A sequential search through all pixels of the larger image could take longer than what is required to obtain the necessary information. Thus, the use of an optimized search technique should accelerate the location of the target in the larger image. The combined usage of the dedicated hardware to compute the required PCCs with and optimized software-based search makes it possible to locate and track targets in images in videos at 30 frames per second.

Among the various optimized search techniques, those of swarm intelligence are based on real-world social behaviors, based on interaction and organization of computational agents of simple resources to perform tasks [14,15,16]. Agents interact with the environment and cooperate with each other in an attempt to produce solutions to complex problems, exploring points in the vicinity of others previously classified as good. A common limitation of these techniques is the adequate choice in parameters, which directly impacts the processing time to reach the global optimum, or even the impact on the susceptibility of converging to local optima, providing false positives and impairing decision making.

This work proposes to implement TM in a co-design system and investigate six different swarm intelligence techniques to accelerate the search process. These techniques are based on behavior of cuckoo birds, bees, elephants, bacteria, fireflies, and fireworks. In addition, this study implements the use of a coprocessor. For each technique, the parameters are configured to process video images at 30 frames per second. To evaluate the performance of each technique, the average processing time, the average number of iterations, the average success rate, and their respective standard deviations are considered.

The proposed system can operate according to three different approaches: the first uses only an FPGA-based general-purpose processor for the search; the second uses a coprocessor operating in serial mode to compute the required PCCs; the third uses a coprocessor operating in parallel mode, via a pipeline, to process the images even faster. Furthermore, each search technique is implemented using the programming environment of the FPGA-based board Smart Video Development Kit, which is used to prototype the proposed co-design tracking system, considering the three aforementioned approaches. Mainly, the contribution of this work consists of an effective, concise, yet efficient real-time hardware system for object tracking that uses swarm intelligence techniques to assist during the tracking process. The proposed design can be embedded into any bigger device.

The remainder of this paper is organized into six sections. Initially, in Section 2, we present the details in the problem of object tracking, wherein emphasis is assigned to TM, the technique adopted in this work to identify the target. Then, in Section 3, we present the works related to object tracking in images and videos. After that, in Section 4, we describe the co-design system to efficiently implement the proposed object tracking solution, wherein emphasis is assigned to the hardware subsystem architecture. In Section 5, we present the optimized swarming search techniques implemented in this work. Later, in Section 6, we describe the methodology adopted during the experiments and present the results of the implementation of the optimized search strategies used to implement the software subsystem, including a thorough comparative study regarding the performances of the techniques. Finally, in Section 8, we draw pertinent conclusions about which search strategy proved to be more suitable for searching and tracking targets in images, depending on the prescribed performance goals. We also suggest some directions for future work improvement.

## 2. Pattern Tracking Problem

A pattern is a collection of objects that are similar to each other, arranged in a way that is in contradiction of their natural arrangement [17]. It can also be defined as the opposite of chaos, an entity, loosely defined, which one can assign a specific name [18]. For pattern tracking, tracked objects are usually called patterns [19]. Objects can be defined as something of interest for future analysis. For example, in images, tracking boats at sea, vehicles on the road, aircraft in the air, and people walking on the street can be considered monitoring for a certain purpose and thus tracking [20].

Pattern recognition is one of the most important and active branches of artificial intelligence. It is the science that tries to make machines as smart as human beings in recognizing patterns, among the desired categories, in a simple and reliable way [21,22]. It is also defined as the study of how machines can observe the environment, distinguish various patterns of interest, and make rational decisions. Pattern recognition provides solutions to problems in the most diverse areas such as image analysis, industrial automation, computer vision, biometric identification, remote sensing, voice recognition, face recognition, surveillance, and defense, among many others. Recognizing patterns in images and tracking their positions in videos has been the subject of several studies and has stood out for being a demanding area of image processing and computer vision [23,24]. In this section, we introduce the basic concepts associated with tracking patterns in images.

### 2.1. Pattern Detection

Any tracking method requires a mechanism that can identify the object the first time it appears in the video and also in each frame. The most common approaches used for this purpose are based on segmentation, background modeling, point detection, and supervised learning.

Segmentation partitions the image into similar regions to obtain the object of interest. Segmentation algorithms have to balance criteria for good and efficient partitioning. Some examples of algorithms used for segmentation include *graph-cut* [25] and *active contours* [26]. Background modeling builds a representation of the scene and performs object detection based on the deviations observed in each frame [20]. Scene objects are classified by forming a boundary between the background and the foreground. The foreground contains all objects of interest. Some examples of algorithms used for background modeling include *background subtraction* and *frame differencing*, mixture of Gaussian functions [27], *eigenbackground* [28] and *optical flow* [29].

Point detectors are used to find points of interest in images. These points are called *features* and are highlighted by their distinguishing characteristics in terms of color, texture, geometry, and/or intensity in gradient variation. Object detection is performed by comparing these points. An interesting feature of this approach is its invariance to changes in light and camera position [20]. Some examples of algorithms based on *features* include *Scale Invariant Feature Transform* (SIFT) [30], *invariant point detector* [31], and *Speeded-Up Robust Features* (SURF) [32]. Supervised learning can also be used for object detection. In this case, the task is performed by learning the different points of view of the object, from a set of samples and a supervised learning mechanism. This method usually requires a large collection of samples regarding each class of objects. In addition, the samples must be manually labeled, a time-consuming and tedious task [9]. The selection of the characteristics of the objects in order to differentiate the classes is also an extremely important task for the effectiveness of the method. After learning, the classes are separated, as best as possible by hyper-surfaces in the feature space. Some methodologies using this approach include neural networks [33], *adaptive boosting* [34], and decision tree [35].

It is noteworthy to point out that object detection and tracking are very close and related processes because tracking normally starts with object detection, while repeated object detection in subsequent frames is required to help perform tracking [23].

In order to track an object and analyze its behavior, it is essential to classify it correctly. The classification is directly linked to the characteristics of the object and how it is represented. Approaches to classification are often based on the object’s shape [36], movement [29,37], color [23], and texture [38,39].

### 2.2. Tracking Techniques

Tracking can be defined as a problem of approximating the trajectory of an object in a given scene. The main purpose is to find the trajectory of this object by finding its position in each video frame [37]. Basically, tracking techniques can be divided into the following categories: point-based tracking, kernel-based tracking, and silhouette-based tracking. Figure 1 illustrates the three categories for camera tracking in the known “Cameraman” image. The tasks of detecting the object and matching those of the previous and subsequent frames can be performed together or separately [9].

#### 2.2.1. Point-Based Tracking

For point-based tracking, the objects are represented by dots and the position of the dots in the frame sequence allows the tracking to occur. This approach requires a mechanism to detect the objects in each frame. The Kalman filter, which is a recursive algorithm that provides a computationally efficient means of estimating the system state, is usually used to estimate the position of objects, based on the dynamics of movement along the video. A limitation of the Kalman filter is the assumption that the variables are normally distributed. Thus, when the state variables do not follow a Gaussian distribution, the estimate does not produce good [9] results. This limitation can be overcome with the particle filter, which uses a more flexible state space model. Multiple Hypothesis Tracking (MHT) is another method which is generally used to solve multiple target tracking problems. It is an iterative algorithm based on predefined assumptions about the object trajectories. Each hypothesis is a set of disconnected trajectories. For each hypothesis, the estimate of the target in the next frame is obtained. This estimate is then compared to the current measurement using a distance measurement. This algorithm can deal with occlusions and has the ability to create new trajectories for objects that enter the scene and finalize those related to objects that disappear from the scene.

#### 2.2.2. Kernel-Based Tracking

In pattern tracking, a kernel refers to an object with a notable region related to its shape and appearance. It can be a rectangular area or an elliptical shape. Objects are tracked by the location after their movements, starting from the embryonic region represented by the kernel, from one frame to the next. These movements are usually represented by affine transformations such as translation, rotation, and scaling. Some of the difficulties of this approach are that kernel does not cover the entire procured object and it includes background contents. The latter is usually mitigated by the layering-based technique, which models the image as a set of layers. One layer is associated with the background and the others are associated with each object in the image. The probability of each pixel belonging to a layer (object) considers the shape characteristics and previous movements of the object. This method is generally useful to track multiple objects.

Template matching, also known as model matching, is a brute force method that looks for regions of the image that are similar to a reference image that represents the procured object, called the template. The position of the template in the image is computed from similarity measures, such as sum of absolute differences, sum of squared differences, cross-correlation, and normalized cross-correlation, among others. This method is capable of handling single-image tracking and background changes. A limitation of template matching is the high computational cost associated with brute force. Many researchers, in order to reduce this cost, limit the search area to the neighborhood of the object in the previous frame [9]. We explore this method in this work; it will be further detailed in Section 2.3.

#### 2.2.3. Silhouette-Based Tracking

Objects can have complex shapes that cannot be well described with simple geometric shapes [9]. Silhouette-based tracking methods aim to identify the precise shapes of objects in each frame. This approach can be divided into two categories, depending on how the object is tracked: by contours or by shapes. *(i)* Contour matching approaches evolve the initial contour of the object to its new position. It is necessary that part of the object in the previous frame overlaps with the object in the next one. There are many algorithms that extract object contours, such as the one called active contours (or *snakes*), based on the deformation of the initial contour at determined points [40]. The deformation is directed towards the edges of the object by minimizing the snake energy, pushing it towards lines and edges. *(ii)* Shape matching approaches are very similar to template matching. The main difference is that the model represents the exact shape of the object. An example of this type of method is presented in [41]. The algorithm uses the Hausdorff distance to find the location of the object.

### 2.3. Template Matching

Template matching (TM) is widely used in image processing to determine the similarity between two entities of the same type (pixels, curves, or shapes). The pattern to be recognized is compared with a previously stored model, taking into account all possible positions. The task basically boils down to finding occurrences of a small image, considered the template, in a sequence of larger images of the frames. Figure 2 shows two matrices representing two black and white images. The image in Figure 2b represents the *template* to be found in the image of Figure 2a. In integer-byte representations for black and white images, the larger the value of a pixel, the closer to white it is, and the smaller the value of the pixel, the closer to black it is.

The search in the frame is conducted by comparing the template, in each pixel, with pieces of image of the same size. The template slides, pixel by pixel, on the main image until all positions are visited. At each position, a similarity measure is computed and used to compare the images. After calculating all similarity measures, the one with the highest value, above a pre-established threshold, is considered to be the location of the sought template within the frame [42]. This operation is very costly when considering large models and extensive sets of frames [21]. The advantage of template matching is that the template stores several particular characteristics of the object (color, texture, shape, edges, centroid, etc.) which differentiate it from others, allowing greater accuracy and tracking of a specific object within a group of similar ones. Furthermore, object detection is not compromised by choosing how to classify or represent it. The disadvantage is the high computational cost required for the computation of the similarity measure at all image pixels.

To evaluate the degree of similarity of the template along the frame, a range of techniques are used. These include the sum of absolute differences (SAD), sum of squared (SSD), and cross-correlation (CCO). For a given patch, i.e., original image patch A of the same size as the procured template, these indices are computed as shown in Equations (1), (2), and (3), respectively: (1)$$\begin{array}{ccc} \mathrm{SAD} & = & \sum_{i=1}^{N}\left|\left(p_{i}-a_{i}\right)\right|; \end{array}$$(2)$$\begin{array}{ccc} \mathrm{SSD} & = & \sum_{i=1}^{N}{\left(p_{i}-a_{i}\right)}^{2}; \end{array}$$(3)$$\begin{array}{ccc} \mathrm{CCO} & = & \sum_{i=1}^{N}p_{i}a_{i}, \end{array}$$
where *N* is the overall number of pixels in the template and patch, $p_{i}$ is the intensity of pixel *i* in the template image, and $a_{i}$ is the intensity of pixel *i* in patch *A*.

Note that in the case of the similarity metrics SAD and SSD, the closer to zero the index is, the more similar the compared images are. However, CCO is sensitive to changes in the amplitude of images’ pixels [43]. To overcome this drawback, normalized cross-correlation (NCC) is used. It is noteworthy to point out that, in this work, we use NCC, which we explain in detail hereafter.

The term correlation is widely used in common language to mean some kind of relationship between two things or facts. In the field of signal processing, cross-correlation is obtained by the convolution of one signal by its conjugate. In this work, the term correlation has a more restricted meaning and refers to the similarity measure associated with the normalized cross-correlation between two images. This metric is an improved version of simple cross-correlation CCO. It features a normalizing value in the denominator that provides it invariance to global changes in brightness and results always within the range [−1,1]. The normalized cross-correlation, also known as Pearson’s correlation coefficient (PCC) [44], is defined in Equation (Equation 4):(4)$$\mathrm{PCC}=\frac{\sum_{i=1}^{N}\left(p_{i}-\bar{p}\right)\left(a_{i}-\bar{a}\right)}{\sqrt{\sum_{i=1}^{N}{\left(p_{i}-\bar{p}\right)}^{2}}\sqrt{\sum_{i=1}^{N}{\left(a_{i}-\bar{a}\right)}^{2}}},$$
where $p_{i}$ is pixel intensity *i* in the template image; $\bar{p}$ is the average pixel intensity of the template image; $a_{i}$ is the intensity of pixel *i* in patch *A*; and $\bar{a}$ is the average intensity of the pixels in patch *A*. The template and patch A must be the same size, and the overall number of pixels is *N*.

The PCC can be understood as a dimensionless index with values between −1 and +1, inclusive, which reflects the intensity of the degree of the relationship between the two compared images. A coefficient equal to 1 means a perfect positive correlation between the two images. A coefficient equal to −1 means a perfect negative correlation between the two images. A coefficient equal to 0 means that the two images do not linearly depend on each other.

The ideal use of the normalized cross-correlation, presented in Equation (Equation 4), considers that the appearance of the target remains the same throughout the video [45]. It is noteworthy to mention that any change in target scale or rotation can influence metric values. Additionally, the change in lighting conditions and/or noise, also known as clutter, that is inserted into the environment can cause errors. A possible solution to this problem is to update the template at every frame, allowing adaptive correlation.

## 3. Related Works

In [13], different TM methodologies were analyzed for their performance regarding variations in illumination, contrast, and position of observation of the target in an image. Some works analyzed sought to increase robustness in relation to the ability to maintain efficiency in the presence of these variations. Area-based and feature-based methods have also been described. A matched filter system was proposed to increase the signal-to-noise ratio to facilitate target identification [46]. In contrast with the present work, there was no application of a noise-removal filter, but this could improve the ability to detect targets in the sequence of frames of a video. In [12], evaluation methods for TM were presented based on CCO and PCC. TM was used to classify objects, aiming to find a small image (*template*) within a larger one or to identify similar images. In contrast, TM and PCC were employed in this work to find the target in each frame. In [47], a TM application of real-time tracking was proposed. Therein, a fast and compressed tracking methodology was presented, in which a minimal computational cost was employed, combined with a high frame rate per second in the video. TM normalized cross-correlation was incorporated to increase the robustness of the system due to the need to work with real-time processing. Bayesian classifiers, correlation filters, similarity measures, and particle filtering were used to determine the most likely location of the target in the frames.

In [48], a combination of the Cuckoo Search technique with a particle filter [9,49] was applied to solve the tracking problem. The search technique introduced some randomness in the process. Another important contribution of the work was a new way to address scaling and rotation errors during tracking. Similarly, in [50], an implementation of the CS combined with the Kalman filter was proposed and applied to tracking. An implementation of TM with PCC was presented in [51] using the PSO, exhaustive search, and GA techniques. The TM technique was used to look for the similarity between a template and the image patches based on PCC. The results quantified PSO’s performance up to 131 times better with the exhaustive search.

An autonomous navigation system based on identifying landmarks in images using a Kalman filter was proposed in [52]. This system utilizes recognition and tracking processes for decision making, where the distances to the references are estimated, as well as the angles between the vehicle trajectory and a fixed reference platform. In [53], a particle filter implementation in FPGA was presented for the identification of objects in environments that are nonlinear and non-Gaussian. A parallel pipelined implementation was proposed used to improve the system’s computational efficiency.

In [54], an implementation on the Xilinx Zynq-7000 FPGA platform, dual-core ARM processor, and NEON vector coprocessor was tailored for the detection and tracking of moving objects. The system’s algorithm is based on the dynamic differences in the image background. In [7], a solution to the tracking problem is proposed through a hardware/software co-design system using FPGA and applying a particle filter. The objective was to monitor the movement of customers inside stores. The work focused on images that are susceptible to variation in template lighting and occlusion.

There are many works studying object detection and tracking based on machine learning. For the sake of generality, we comment on some of the recent literature on the subject. Therein, optimization techniques may be exploited in the process. However, it is noteworthy to point out that these works are not concerned with the hardware implementation of the tracking methodology. In [1], a learning strategy based on two swarm-intelligence-based techniques, namely, PSO and BFOA, was proposed. It aimed to optimize the parameters of the classifier and loss function of a Region Proposal Network, which was specifically designed for object detection to improve feature-sampling ability. The tuned network was used in an autonomous driving application. In [55], the authors propose an algorithm for vehicle detection and classification to overcome the problems regarding complex scenes with backgrounds and objects of reduced size in large scenes. The algorithm is based on a convolutional neural network to classify the detected vehicle into two classes: light and heavy vehicles. In [56], a tracking system, where the tracked object is indicated interactively, was designed. It involves a tracking process that combines a contour detection network and a fully convolutional Siamese tracking network. The authors showed that the system can be used in real-time tracking.

In [57], an efficient and accurate vehicle detection algorithm in aerial infrared images was proposed via an improved YOLOv3 network. To increase the detection efficiency, the anchor boxes were increased 4× to improve the detection accuracy of the small vehicles. In [58], addressing the shortcomings of the current YOLOv3 model, such as large size, slow response speed, and difficulty in deploying to real devices, a new lightweight target detection network called YOLOv3-promote was proposed. The experimental results show that the inference speed of the proposed methodology is about 5 times that of the original model.

## 4. Proposed Co-Design System

The macro-architecture of the proposed integrated system is shown in Figure 3 and includes a general-purpose processor for executing the PSO step, a coprocessor for calculating the PCC, dedicated memory blocks (BRAM IMG and BRAM TMP) to store the main image and template, respectively, and access control blocks to these memories (GET IMG and GET TMP). The components are described using the hardware description language VHDL (*Very-high-speed integrated circuits Hardware Description Language*) and synthesized using the software tool Vivado, from Xilinx. The integrated system developed in Vivado can be seen in Figure 4.

Figure 5 presents the proposed architecture for the coprocessor. It is responsible for yielding the correlation between two images, as defined in Equation (Equation 4). It is designed to operate in a pipeline, where each of its three blocks corresponds to one of its three stages. At each rising transition of the clock signal, three data points are received by the coprocessor:data_p: a pixel of the reference image (template), represented by 8 bits;data_ac: a pixel of the image to be compared, also represented by 8 bits;data_am: a pixel of the next image to be compared, of 8 bits.

The images to be compared have a size of 64 × 64 pixels, a total of 4096 pixels. This is represented by 4 KBytes. The computations are performed in each block and passed to the next one at each synchronization pulse. This pulse is generated by Block Synchro at the 4103rd rising transition of the clock signal. As output, the coprocessor provides the obtained correlation value (result), in 2’s complement, encoded in 32 bits, a flag signal (done) indicating completion, and an error signal (error) indicating that the result is not valid when a division by zero has occurred.

Figure 6 shows the micro-architecture of Block 1 that forms the first stage of the pipeline. It is responsible for computing the average value of the pixels of the two compared images. It has two output registers that are loaded only when the stage task is completed. Every clock pulse, component AVG is restarted.

Figure 7 shows the micro-architecture of Block 2 that forms the second stage of the pipeline. It is responsible for computing the three summations of the normalized cross-correlation of Equation (Equation 4). It is composed of two components SUB that perform the subtraction in 2’s complement of the pixels of the images with the averages returned by Block 1; three components MULT, which perform, in one clock pulse, the multiplication of the results returned by components SUB; and the components SUM, which carry out the sums, in 2’s complement, of the results of the multiplications returned by components mult. As in the case of Block 1, Block 2 has two output registers that are loaded only when the stage computation is completed. Every clock pulse, components SUB, MULT, and SUM are restarted.

Figure 8 shows the architecture of Block 3, which composes the third stage of the pipeline. It is responsible for computing the main multiplication, square root, and division of the normalized cross-correlation coefficient, as defined in Equation (Equation 4)). The pipeline stage includes component MULT, which, in a single clock pulse, performs the multiplication of the sums of the denominator of Equation (Equation 4); and component SQRT, which yields the square root and component PDIV, which performs the division, producing a precise factional result. The quotient is assigned $\pm 2^{-24}$. This last component is the one that provides the coprocessor output signals. The operation of this block is controlled by the state machine FSM, coordinating the actions of the included components. At every synchronization pulse, the machine returns to its initial state. Like Block 1 and Block 2, it includes output registers that are loaded only when the stage task is completed.

Based on this numerical method called *babylonian* [59], component SQRT of Block 3 is implemented, in the hardware, using an iterative process. Furthermore, Algorithm 1 is used to implement, in the hardware, the accurate division upon which component PDIV is based.
**Algorithm 1** Accurate division of Q=A/B with 24 bits$N_{B}$: = 0; $RA_{\langle N-1\cdots \frac{N}{2}\rangle }$: = A; $RA_{\langle \frac{N}{2}-1\cdots 0\rangle }$: = 0**while** $RA_{\langle N-1\cdots \frac{N}{2}\rangle }>B$ **do**   RA: = right-shift RA; $N_{B}$: = $N_{B}+1$**end while***Q*: = 0**for** $j=1,N_{B}\;$**do**   RA: = left-shift RA; *Q*: = left-shift *Q*   **if** $RA_{\langle N-1\cdots \frac{N}{2}\rangle }>B$ **then**     $RA_{\langle N-1\cdots \frac{N}{2}\rangle }$: = $RA_{\langle N-1\cdots \frac{N}{2}\rangle }-B$; $Q_{0}$: = 1   **else**     $Q_{0}$: = 0   **end if****end for****for** j=1,24 **do**   RA: = left-shift RA; *Q*: = left-shift *Q*   **if** $RA_{\langle N-1\cdots \frac{N}{2}\rangle }>B$ **then**     $RA_{\langle N-1\cdots \frac{N}{2}\rangle }$: = $RA_{\langle N-1\cdots \frac{N}{2}\rangle }-B$; $Q_{0}$: = 1   **else**     $Q_{0}$: = 0   **end if****end for**

The dedicated memory blocks BRAM_TEM and BRAM_IMG store the template and the main image, respectively. These are implemented in the programmable logic of the Zynq chip (PL). Memory BRAM_TEM can store up to 4096 8-bit pixels, summing up 4K bytes, which corresponds to the size of template. Memory BRAM_IMG can store up to 573 × 463 pixels of 8 bits each, summing up 260K bytes. As the edges of the main image are padded with zeros, the maximum size of this image is thus 510 × 400 pixels. Components GET_TEM and GET_IMG are responsible for providing access to the dedicated memories BRAM_TEM and BRAM_IMG, respectively. They make the data available to the coprocessor at the right time. The access processes of reading and writing to the memories are synchronized by the clock signal (CLK) and by the synchronization signal (CLK_sync).

## 5. Swarm-Intelligence-Based Search Strategies

This section presents the six optimized search techniques applied in this work to the object tracking problem. The respective canonical algorithms are presented. In Section 5.1, we present the technique inspired by the reproductive behavior of the cuckoo bird. Then, in Section 5.2, we analyze the optimized search method based on the behavior of bee colonies in the search for food. After that, in Section 5.3, we present the technique inspired by the social behavior of elephants in their herds. Subsequently, Section 5.4 describes the optimization technique that seeks to mimic the behavior of bacteria *E. Coli* in the search for food, movement, reproduction, and their own survival in the environment they inhabit. Section 5.5 introduces the technique inspired by the behavior of fireflies. In Section 5.6, we address a technique modeled on the movement and dissemination of incandescent particles from fireworks during their detonation. In Section 5.7, we present a search technique used as a performance reference.

### 5.1. Cuckoo-Search-Based Technique

Cuckoo Search is an optimization algorithm that is population-based meta-heuristic, and is inspired by the behavior of cuckoos in nature. A swarm of candidate solutions, which are identified as nests, is used to search for the optimal solution to a problem. The algorithm uses a combination of random walk and Lévy flights to explore the search space, and the best solutions are selected for future swarms. The algorithm also employs a mechanism that mimics the the process of laying eggs by cuckoos. Therein, a fraction of the nests are randomly replaced with new ones in order to maintain diversity in the solution represented by the swarm. Cuckoo Search can be applied to solve any optimization problem, including function optimization, multi-objective optimization, and combinatorial optimization problems. In general, it shows the ability to escape from local optima, which makes it an efficient choice for solving optimization problems. Overall, the Cuckoo Search algorithm is a robust and efficient swarm-intelligence-based technique that can be used to find near-optimal solutions to complex problems. The CS main steps, used in this work, are described in Algorithm 2. Therein, the parameters to be adjusted according to the application are: Lévy step scale factor *L*; number of cuckoos $N_{c}$; maximum number of iterations *M*; probability of nest discovery $P_{a}$; and stopping criterion, which allows the process to stop as soon as it finds the sought solution or reaches the generation limit.
**Algorithm 2** Cuckoo Search main stepsDefine objective function f(x)Define parameters *L*, $N_{c}$, *M*, $P_{a}$Generate initial positions $x_{i}$ of the host nests*t*: = 1**while** (t≤M) and (solution not yet found) **do**   Choose cuckoo *j* randomically   Move it using a Lévy step   Evaluate the quality of the new nest using f(x)   Sort existing nests based on the quality   Abandon
$P_{a}$ of the worst nests   Generate new ones around the best nest   Keep the remaining 1 − $P_{a}$ nests with the best quality   Sort nests and pick the best one   *t*: = *t* + 1**end while**Return the best nest

### 5.2. Artificial-Bee-Colony-Based Technique

The Artificial Bee Colony (ABC) algorithm is an optimized search technique proposed in [60] and inspired by the foraging behavior of bees. The positions occupied by food sources in the search space represent possible solutions to the problem, and the amount of nectar available in the source is associated with the quality of the solution, defined through the objective function. This technique presents the mimicking of communication activities, task allocation, swarm location selection, mating and reproduction, nectar search, and pheromone diffusion. Three types of bees are defined: employed bees, onlookers, and scouts. Employed bees carry routes to food sources and inform other bees about the quality of the source. Onlookers wait for new information from the employed bees in the hive. Scouts search for new food sources in the environment, independently of other known sources. The number of solutions is the sum of employed bees and onlookers. The ABC main steps, used in this work, are described in Algorithm 3. Therein, the parameters to be adjusted according to the application are: $N_{a}$ of bees (onlookers), which is equal to the number SN of food sources; *D*, which is the threshold value of the search space; MCN, which is the maximum number of cycles; $N_{exh}$, which is the number of iterations that defines whether a source *i* is exhausted; and $E_{exp}$, which defines the neighborhood to be considered at when looking for new food source.
**Algorithm 3** ABC main stepsInitialize the food source locations (solutions) $x_{ij}$, i=1,…,SN e j=1,…,DDefine $N_{exh}$ e $E_{exp}$; *t*: = 1**while** (t≤MCN) and (solution not yet found) **do**   Compute $v_{ij}=x_{ij}+\phi_{ij}\cdot (x_{ij}-x_{kj})$   Evaluate new solutions using objective function   Apply selection greedy process   Compute probabilities $P_{i}=\frac{f(x_{i})}{\sum_{1}^{SN}f(x_{n})}$   Compute onlookers $v_{ij}$ using selected $x_{ij}$ and $P_{i}$   Evaluate new solutions using objective function   Apply selection greedy process   Determine source to abandon when hitting $N_{exh}$   Replace the abandoned source $x_{i}^{j}=x_{best}^{j}+rand\left(0,1\right)\cdot \left(x_{max}^{j}+x_{min}^{j}\right)$   Store the best solution (food source) found so fa   *t*: = *t* + 1**end while**Return best solution

### 5.3. Elephant-Herding-Based Technique

Elephant Herding Optimization (EHO) [61] is a technique based on swarm intelligence and inspired by the social behavior that occurs in herds of elephants. Elephants are social animals with complex structures comprising females and young. A group of elephants is composed of many clans under the command of a matriarch, typically the oldest one. The clan consists of a female with her offspring or with some other related females. Female elephants prefer to live in families, while males tend to live in isolation, which leads them to leave their families when they grow up. When they wander off, they can maintain contact with elephants from other clans through communication via low-frequency vibrations they produce by hitting their feet against the ground. Elephants belonging to different clans live together under the leadership of a matriarch. Male elephants leave their family groups when they grow up. These two behaviors can be modeled through two operators: clan updating and separation.

In EHO, elephants of each clan are updated, taking into account the current position and the matriarch during the action of the clan updating operator. Then, the separation operator can increase the population diversity in the final search phase. The EHO main steps, used in this work, are described in Algorithm 4. Therein, the parameters to be adjusted according to the application are: $N_{e}$, which represents the number of elephants, and $NG_{max}$, which denotes the maximum number of cycles.

### 5.4. Bacterial-Foraging-Based Technique

The optimized search technique based on the behavior of colonies of the bacteria E. Coli, called Bacterial Foraging Optimization Algorithm (BFOA), was introduced in [62]. The positions of the bacteria in the search space represent the solutions, which are evaluated through the previously defined objective function. During the movement of the bacteria, the positions tend to evolve until the global optimum of the function is reached. In addition, they reproduce, act collectively in their movements, and are influenced by the facilitating or hindering effects of their environment. The foraging behavior of *E. coli* bacteria is synthesized through the following processes [62,63]: chemotaxis, swarm behavior, reproduction, elimination, and dispersal. The combination of these processes models the way bacteria survive in their environment, which can be more favorable (having adequate food) or repellent (containing toxic substances for the bacteria). The bacteria are able to perceive this gradient in the environment and make decisions on where to go. The BFOA main steps, used in this work, are described in Algorithm 5.
**Algorithm 4** EHO main stepsDefine $N_{e}$ and $NG_{max}$*t*: = 1**while** ($t\leq NG_{max}$) and (solution not yet found) **do**   **for** $c_{i}$: = $1,\cdots ,N_{clan}$ **do**     **for** *j*: = $1,\cdots ,N_{c_{i}}$ **do**        Update $x_{c_{i},j}$        Generate $x_{new,c_{i},j}=x_{c_{i},j}+\alpha \cdot \left(x_{best,c_{i}}-x_{c_{i},j}\right)\cdot r$        **if** $x_{c_{i},j}=x_{best,c_{i}}$ **then**          Update $x_{c_{i},j}$          Generate $x_{new,c_{i},j}$          Compute $x_{center,c_{i}}=\frac{1}{n_{c_{i}}}\cdot \sum_{j=1}^{n_{c_{i}}}\left(x_{c_{i},j}\right)$          Generate $x_{new,c_{i},j}=\beta \cdot x_{center,c_{i}}$        **end if**     **end for**   **end for**   **for** $c_{i}$: = $1,\cdots ,N_{clan}$ **do**     Replace worst solution $x_{worst,c_{i}}=x_{min}+\left(x_{max}-x_{min}+1\right)\cdot rand$   **end for**   *t*: = *t* + 1**end while**Return best solution

**Algorithm 5** BFOA main steps
Define f(x), $NG_{max}$Define $N_{b}$, $P_{ed}$, $N_{ed}$, $N_{rep}$, $N_{chemo}$, $N_{sw}$ e $K_{step}$Generate initial position $x_{i}$ of the $N_{b}$ bacteria*t*: = 1**while** ($t\leq NG_{max}$) and (solution not yet found) **do**   **for** (*l*: = $1,\cdots ,N_{ed}$) **do**     **for** *k*: = $1,\cdots ,N_{rep}$ **do**        **for** *j*: = $1,\cdots ,N_{chemo}$ **do**          Compute f(i,j,k,l) and Compute em $f_{prev}$ with associated $m_{1},m_{2},m_{3}$          Generate mov(i) of random values in [−1,1]          Move bacteria using $f_{i,j+1,k,l}\left(m_{1},m_{2},m_{3}\right)=f_{i,j,k,l}\left(m_{1},m_{2},m_{3}\right)\;+\;C\left(i\right)\cdot \frac{mov(i)}{\sqrt{mov^{T}\left(i\right)\cdot mov\left(i\right)}}$          **for** *s*: = $1,\cdots ,N_{sw}$ **do**             Move for $N_{sw}$ steps and generate $f_{dev}$          **end for**          **if** $f_{dev}\;>\;f_{prev}$ **then**             Goback to positions $f_{dev}$ for bacteria             **for** *s*: = $1,\cdots ,N_{sw}$ **do**               Move for $N_{sw}$ steps             **end for**          **end if**        **end for**        Reproduce using the best half of $N_{b}$ the bacteria     **end for**     Eliminate $P_{ed}$ of the worst bacteria     Generate the same fraction randomly   **end for**   *t*: = *t* + 1
**end while**
Return best position


In BFOA, the parameters to be adjusted according to the application are: the number of bacterial generations $NG_{max}$; the amount of bacteria $N_{b}$; the number of steps of elimination and dispersion $N_{ed}$, of reproduction $N_{rep}$, and of chemotaxis $N_{chem}$; the probability $P_{ed}$ of elimination and dispersion; the number $N_{sw}$ of swimming movements of each bacterium; and the size $K_{step}$ of the swim.

### 5.5. Firefly-Behavior-Based Technique

The Firefly Algorithm (FFA) is a technique inspired by the behavior of fireflies, proposed in [64] for optimization problems with constraints. The algorithm is based on the bioluminescent signals, i.e., lights emitted by insects, used for communication between fireflies and to repel predators. Fireflies exhibit swarm intelligence characteristics for self-organization and decentralized decision making. The luminous signals are important not only in the search for food, but also in the mating ritual of these insects. In [65], the author reported that one advantage of FFA is its ease of use with other algorithms in a hybrid manner to improve performance. The rhythm of the emissions, their frequency, and their duration define a communication that attracts both sexes. Depending on the species, females may emit a unique response pattern or imitate the patterns of other species to seduce, attract, and then devour males. The perception of light intensity by fireflies is inversely proportional to the distance between the emitter and the receiver, so their communication is affected by distance. This occurs due to light absorption by the air, which is quantified by an absorption coefficient. The brightness intensity of the light emitted by a male firefly is also an indicator of its fitness. However, the FFA technique considers the insects to be unisex, and assigns the attractiveness of the firefly based on the brightness intensity. The intensity refers to the value along the objective function of the position. Initially, a population of fireflies is created. If one of the parameters that influence evaluation is changed, the fitness of the fireflies is recalculated, and they are sorted according to it. The best insects are kept for the next round of evaluation. The FFA main steps, used in this work, are described in Algorithm 6. Therein, the parameters to be adjusted according to the application are: the number of fireflies $N_{v}$, the air–light absorption coefficient γ, the number of dimensions *D*, the maximum number of generations $N_{max}$, and the stopping criterion.
**Algorithm 6** FFA main stepsDefine: $N_{v}$, *D*, f(x), γ, *N*Generate firefly positions $x_{i}$, $i=(1,\ldots ,N_{v})$Compute emission light intensity $I_{i}$ using $f(x_{i})$;*t*: = 1**while** ($t\leq NG_{max}$) and (solution not yet found) **do**   **for** *i*: = $1,\cdots ,N_{v}$ **do**     **for** *j*: = $1,\cdots ,N_{v}$ **do**        **if** ($I_{j}\;>\;I_{i}$) **then**          Move firefly *i* towards *j*        **end if**        Compute firefly perceived attraction        Evaluate solution via f(x)        Update light intensity     **end for**   **end for**   Sort fireflies;   *t*: = *t* + 1**end while**Return best solution

### 5.6. Firework-Behavior-Based Technique

The Fireworks Algorithm for Optimization (FWA), proposed in [66], models the incandescent particles (sparks) from firework explosions and is used in complex optimization problems. The FWA main steps, used in this work, are described in Algorithm 7. Therein, the parameters to be adjusted according to the application are: the number of sites for firework detonation is $N_{f}$; *a* controls the amount of sparks in the case of a bad explosion; and *b* controls the amount of sparks in the case of a good explosion.
**Algorithm 7** FWA main stepsDefine: $N_{f}$, *a*, *b*, *m*, ξ, f(x), *T*Select randomically $N_{f}$ sites for fireworks detonation;*t*: = 1**while** ($t\leq NG_{max}$) and (solution not yet found) **do**   Detonate the $N_{f}$ fireworks in the selected locations with $s_{i}=m\cdot \frac{\xi +y_{worst}-f\left(x_{i}\right)}{\xi +\sum_{i=1}^{n}\left(y_{worst}-f\left(x_{i}\right)\right)}$ sparks each   **for** *i*: = $1,\cdots ,N_{f}$ **do**     **if** $s_{i}<a\cdot m,$ **then**        Compute ${\hat{s}}_{i}=\left\lceil a\cdot m\right\rceil$     **else if** $s_{i}$>b·m, a<b<1,        Compute ${\hat{s}}_{i}=\left\lfloor b\cdot m\right\rfloor$     **else**        Compute ${\hat{s}}_{i}=\left\lfloor s_{i}\right\rfloor$     **end if**     Yield location of ${\hat{s}}_{i}$ sparks $x_{i}$ using the 5 steps:       Initiate a spark location ${\hat{x}}_{j}=x_{i}$       Compute z=⌈d·r⌉       Choose randomically *z* dimensions of ${\hat{x}}_{j}$       Compute displacement $h=A_{i}\cdot r$       **for** each dimension ${\hat{x}}_{k}^{j}\in z$ **do**              ${\hat{x}}_{k}^{j}={\hat{x}}_{k}^{j}+h$              **if** $\left({\hat{x}}_{k}^{j}<{\hat{x}}_{k}^{melhor}\right)$ or $\left({\hat{x}}_{k}^{j}>{\hat{x}}_{k}^{pior}\right)$ **then**                       ${\hat{x}}_{k}^{j}={\hat{x}}_{k}^{best}\;+\;\left|{\hat{x}}_{k}^{j}\right|$ % $\left({\hat{x}}_{k}^{worst}-{\hat{x}}_{k}^{best}\right)$              **end if**       **end for**     **end for**     **for** *k*: = $1,\cdots ,\hat{m}$ **do**        Select randomically firework $x_{j}$.        Yield one spark using the 5 steps:       Initiate a spark location ${\hat{x}}_{j}=x_{i}$       Compute z=⌈d·r⌉       Choose randomically *z* dimensions of ${\hat{x}}_{j}$       Compute Gaussian explosion coefficient *g*       **for** each diemnsion ${\hat{x}}_{k}^{j}\in z$ **do**               ${\hat{x}}_{k}^{j}={\hat{x}}_{k}^{j}\cdot g$               **if** $\left({\hat{x}}_{k}^{j}<{\hat{x}}_{k}^{best}\right)$ or $\left({\hat{x}}_{k}^{j}>{\hat{x}}_{k}^{worst}\right)$ **then**                         ${\hat{x}}_{k}^{j}={\hat{x}}_{k}^{best}\;+\;\left|{\hat{x}}_{k}^{j}\right|\;\%\;\left({\hat{x}}_{k}^{worst}-{\hat{x}}_{k}^{best}\right)$               **end if**          **end for**        **end for**        Select the best site and keep for the next iteration        Select randomically $N_{f}-1$ locations for 2 spark types        Select Current fireworks;        *t*: = *t* + 1     **end while**     Return best solution

In the FWA technique, there are two search processes employed that are related to two specific types of explosion. Mechanisms for maintaining the diversity of sparks are applied. When fireworks are set off, a shower of sparks fills the space. In [66], the explosion was interpreted as a search process around a specific point by the sparks produced. In an attempt to find a specific point $x_{j}$ that satisfies $f(x_{j})=y$, fireworks are continuously set off in the search space until a spark hits the desired point or a region considered sufficiently close to it. For each generation of fireworks, $N_{f}$ locates where the fireworks are detonated and selects them. After each explosion, the sparks hit new points, which are evaluated. If the optimal location is found, the algorithm stops. On the other hand, $N_{f}$ selects other locations from the current sparks to generate the next generation. By observing real firework explosions, two specific types were identified to be modeled. If the detonations are correct, a large number of sparks is produced around the center of the explosion. In case of a defective detonation, few sparks are released and scattered around the environment. A good firework detonation in FWA means that the sparks have found a promising area, which may be closer to the desired optimum; thus, it is appropriate to use more sparks to search around the detonation location. At the same time, a poor detonation is related to the fact that the explosion point is far from the desired optimum, which would require a larger search radius.

### 5.7. Particle-Swarm-Based Technique

The Optimized Particle Swarm Optimization (PSO) search technique, proposed by Kennedy in 1995, is inspired by the collective behavior of birds and fish. In PSO, the search space is explored by a defined number of particles that move to find the optimal point of the established objective function.

The particles have a position characterized by a coordinate from each of the objective function dimensions and an individual velocity that is constantly updated based on collective movement and their own experiences. The best position occupied by each particle so far and the best position obtained by the swarm particles up to that point are stored. The quality of the particles in the search space is calculated using the objective function that models the problem, and each point represents a potential solution to be evaluated. The PSO main steps, used in this work, are described in Algorithm 8. Therein, the parameters to be adjusted according to the application are: the number of particles $N_{p}$, the coefficients ω, $\phi_{1}$, $\phi_{2}$, $c_{1}$, and $c_{2}$; the maximum number of generations $NG_{max}$; and the maximum velocity $V_{max}$.
**Algorithm 8** PSO main stepsDefine: ω, $\phi_{1}$, $\phi_{2}$, f(x), *M*Initialize randomically the $N_{p}$ particles in the search spaceInitialize $P_{i_{best}}=f\left(x_{i}\right)$Initialize $S_{best}$ as $G_{b}est=best\left(P_{i_{best}}\right)$ or $L_{best}=best\left(P_{i_{best_{neighbors}}}\right)$*t*: = 1**while** ($t\leq NG_{max}$) and (solution not yet found) **do**   **for** each particle $x_{i}$ **do**     Compute $v_{i}\left(t+1\right)=\omega \cdot v_{i}\left(t\right)+\phi_{1}\cdot r_{1}\cdot \left(P_{i_{best}}-x_{i}\left(t\right)\right)+\phi_{2}\cdot r_{2}\cdot \left(S_{i_{best}}-x_{i}\left(t\right)\right)$     Apply velocity control     Compute new position $x_{i}\left(t+1\right)=x_{i}\left(t\right)+v_{i}\left(t+1\right)$     Update $P_{i_{best}}$ e $S_{best}$   **end for**   Obtain $S_{best}$ among $P_{i_{best}}$ for all particle *i*   *t*: = *t* + 1**end while**Return the position of the best particle

## 6. System Evaluation

The hardware developed is defined as a model for the current work [67], which aims to employ different optimized swarm-intelligence-based search techniques in a co-design approach [68]. It is designed to run either in serial or pipeline mode. The processor available on the board is used to execute each search technique via software, with all the necessary parameter configurations to achieve satisfactory processing time and accuracy rate in finding the target. The performance of the tracking system is evaluated regarding the three compared modes of operation: software only; serial co-design hardware/software; and pipelined co-design hardware/software. We aim to establish the impact of the co-design implementation for the object tracking problem. Therefore, in the rest of this section, we first present a brief description of the board used to implement the co-design system. Then, we describe the dataset used in this performance evaluation. After that, we define the parameters setting used to configure the used swarm intelligence techniques. Finally, we present the performance of the compared techniques and select the best ones.

### 6.1. Implementation Board

The physical equipment used in this work consists of the Smart Vision Development Kit (SVDK) rev 1.2 board. It includes a Xilinx PicoZed 7Z015 System-On-Module (with a Zynq XC7Z015 chip), 1 GB DDR3 memory, and a 33.333 MHz oscillator. The XC7Z015 has a general-purpose processing system based on the dual-core ARM Cortex-A9 processor and programmable logic on a single integrated circuit. The board also features resources such as UART interface, HDMI video encoder, tri-mode Ethernet PHY, general-purpose I/O, USB3, GigE Vision, and CoaXPress interfaces. Its PCI Express bus allows the installation of sensors. The board can be seen in Figure 9.

As explained previously, the proposed architecture features an optimized search technique (CS, ABC, BFOA, EHO, FFA, or FWA) in the software, executed by the processor of the XC7Z015, working in conjunction with a dedicated coprocessor to calculate the PCC. This calculation is the most resource-intensive part of the processing during target localization, so the use of the coprocessor significantly improves performance, a fact proven by the yielded results. In the next step, the SDK tool is used for programming to configure the programmable logic of the XC7Z015. Using the SDK, the C language program is built, which accesses the XC7Z015 and the coprocessor to implement target detection with the aid of swarm intelligence to optimize the search.

### 6.2. Dataset

The tracking system proposed in this work is used to find the targets defined in the images presented in Figure 10, where the original image sizes are also provided. The tested images are part of the VIVID dataset for target image tracking, which is is publicly available [67,69]. The targets selected in the images are highlighted and have 64 × 64 pixels. The search window used is 101 × 101 pixels.

Figure 11 shows the behavior of the objective function, which is the calculation of the PCC, along the *x* and *y* axes for each reference image. The graphs were generated using MATLAB version R2016B. It is possible to observe the presence of local maxima in the images that can attract the swarms during the search, hindering convergence. To avoid the detection of false positives, a minimum threshold of 0.95 was chosen for the objective function, the PCC, so that the target was considered detected.

It is worth emphasizing that the system is evaluated in three configurations:The swarming search technique is implemented in software wherein the XC7Z015’s processor is also used to calculate the PCC whenever required;The search technique is implemented in software while the PCC calculation is performed by the coprocessor, configured in serial mode of operation;The search technique is implemented in software while the PCC calculation is performed by the coprocessor, configured in a parallel mode of operation, using the pipeline.

### 6.3. System Parameter Settings

The goal of this work is to propose a system capable of working with videos at 30 frames per second, which requires finding the target within 33 ms. At the same time, in order for the detection to be considered true, it is necessary to define a minimum PCC threshold. The system’s effectiveness in finding the target requires defining a minimum success rate, measured over sets of 100 searches. The evaluation goals are defined as:Maximum time for target identification: 33 ms;Minimum success rate: 60%.

Each of the three system configurations was applied to the images in Figure 10 and evaluated for processing time, number of iterations required, and success rate obtained. This procedure was repeated 50 times to obtain the mean and standard deviation of the metrics for each set of 100 searches. This resulted in a total of 5000 repetitions to calculate the mean and standard deviation.

The stopping criterion adopted for the code execution was to find the target with PCC ≥ 0.95 or reach a maximum of 10 iterations. The performance goals were verified for the 5000 repetitions of the system execution for each of the seven search techniques, for the three considered configurations.

The target search window within the main image specifies the region where the swarm technique will search for the target. Considering a video, if the target changes its position slowly from frame to frame, the position where it was detected in a given frame will be close to where it will be found in the next frame. Thus, it is not necessary to search the entire image; only the vicinity of the previous location. In this work, a search window with a size of 101 × 101 pixels was defined.

The parameters of each technique were empirically configured to obtain an average success rate of at least 90% via a systematic search of the parameter space. Thus, it is possible to evaluate the relationship between high accuracy and the average processing time required to detect the target in each image. The yielded settings were as follows for the implemented swarming techniques:CS: 50 cuckoos; discovery probability: 25%; Lévy flight scale factor: 15.ABC: 8 bees, which is the same number SN as food sources. $N_{exh}=1$.EHO: 32 elephants; 4 clans; α = 0.75; β = 0.BFOA: 15 bacteria; 1 elimination and dispersal cycle with a probability of 30%; 1 reproduction cycle; 1 chemotaxis cycle; 1 swimming displacement cycle.FFA: 11 fireflies; α = 3; β = 1.6; γ = 0.0005.FWA: number of fireworks $N_{F}$ = 25; *M* = 50; α = 0.02; β = 1.00; ξ = 0.0001.PSO: number of particles $N_{P}$ = 18; ω = 0.6; $c_{1}=0.6$; $c_{2}=2$.

### 6.4. Performance Results

As explained before, for each of the tested techniques, we present the average values and the standard deviations of the the execution time, the number of required iterations to reach the stopping criterion, and the acceptance rate regarding the three system configurations: software only; serial co-design; and pipeline co-design.

Figure 12 regards the performance of the usage of the CS technique. Figure 12a shows the results of the system without using a coprocessor. The processing times obtained were above the necessary threshold (33 ms), which requires system improvement.

The success rate was over 60% for all the images. The best performance was achieved for image $IR3-I_{6}$, which showed the highest average success rate of 97.12%. The shortest average time to find the target was 89.77 ms, while the lowest average number of iterations was 4.53. The worst average time performance occurred when searching for the target in image $Cars-I_{1}$—145.52 ms—which required the highest average number of iterations, 6.48. The worst success rates occurred for images $IR1-I_{4}$, $IR2-I_{5}$, and $Cars-I_{1}$, all above 69%. Figure 12b shows the results for the system configuration with the coprocessor operating in serial mode. The processing time target was achieved for all images, and the success rates obtained were all above 69%. The best performance was achieved when searching for targets located in images $IR3-I_{6}$, $Sedan-I_{3}$, and $Truck-I_{7}$; all had average times below 15 ms and average success rates above 93%. The worst performance occurred when searching for targets in images $Cars-I_{1}$, $IR1-I_{4}$, and $IR2-I_{5}$; the average success rates were below 72.58%. Figure 12c presents the performance of the system when using the coprocessor operating in pipeline mode. The processing times were all below 16 ms; the best system performance occurred for image $IR3-I_{6}$, which had an average time of 12.85 ms and the smallest standard deviation of time among the images. The best average success rate was achieved for image $Truck-I_{7}$: 98.22%The worst average target success rate occurred for images $IR1-I_{4}$ and $IR2-I_{5}$, for which the average success rates were the lowest (67.56% and 72.46%, respectively), and the average numbers of iterations were the highest (7.78 and 5.85, respectively).

Figure 13 regards the performance of the usage of the ABC technique. Figure 13a presents the results of the system without using a coprocessor. None of the average obtained processing times was less than 33 ms. The best average processing time was achieved for image $IR1-I_{4}$ (122.88 ms).

The average success rates were above 70%; only the search in the image $IR2-I_{5}$ did not exceed 80%. The best average success rates occurred for images $Sedan-I_{3}$ and $Truck-I_{7}$, both above 94%. The worst performance was observed in the search for the target in $IR2-I_{5}$, for which the highest average processing time and iterations were required. The same image resulted in the worst average success rate of 79.40%. Figure 13b presents the results for the system configuration with the coprocessor operating in serial mode. There was a significant improvement in the time required to find the target, so that for all images the timing restriction was met, and the lowest average time obtained, of 22.45 ms, occurred for image $IR1-I_{4}$. The obtained average hit rate was over 65%. The search for the target in image $IR2-I_{5}$ required a longer average time of 31.07 ms and more iterations, and it also obtained a lower average hit rate than the searches for targets in the other images. Figure 13c shows the performance of the system with the aid of the coprocessor operating in pipeline mode. There was a slight reduction in the average time and number of iterations; the time target of 33 ms was achieved for all images, achieving average hit rates above 78.32%. The lowest average time was obtained for image $Sedan-I_{3}$: 23.01 ms. The worst performance occurred for the search in image $IR2-I_{5}$, whose average processing time reached 28.76 ms. The number of iterations for this image was also the highest.

Figure 14 regards the performance of the usage of the EHO technique. Figure 14a presents the results of the system without using a coprocessor. As with previous techniques, the obtained processing times were far above the timing restriction of 33 ms, which makes it necessary to improve the system performance.

The lowest average time was obtained for image $IR1-I_{4}$, and the worst for image $IR2-I_{5}$, which showed the highest average number of iterations. The acceptance rate was above 60%, except for the search in image Rcar image, which was 59.10%. The best performances occurred for images $IR1-I_{4}$ and $IR3-I_{6}$; they obtained the lowest average times and highest hit rates. Figure 14b shows the results of the system using the coprocessor in serial mode. There was an improvement in processing time, but not enough to achieve the timing restriction of 33 ms. The best performance in terms of processing time and hit rate occurred during the search for the target in image $IR1-I_{4}$. The worst performance was, again, regarding image $Rcar-I_{8}$, for which the average hit rate, 59.86%, was below 60%. However, it exhibited the highest processing time to find the target, 71.06 ms. Figure 14c shows the performance of the system in the configuration with the use of the coprocessor in pipeline operation mode. The target time of 33 ms was not achieved in any of the images. Once again, the search for the target in image $Rcar-I_{8}$ presented a success rate lower than 60%, and thus it did not reach the minimum performance target.

Figure 15 regards the performance of the usage of the BFOA technique. Figure 15a presents the results of the system without the use of a coprocessor. The obtained processing times did not respect the timing restriction of 33 ms, which again calls for system improvement. The average success rate was higher than 60%, except for the target search in image $IR2-I_{5}$, which was 48.5% due to convergence to local optima. The system performed better in searching for targets in images $IR3-I_{6}$ and $Truck-I_{7}$, respectively.

Figure 15b shows the results of the system with the use of the coprocessor operating in serial mode. There was an improvement regarding processing time, but still with average time values above 33 ms. Once again, the best performances were achieved for images $IR3-I_{6}$ and $Truck-I_{7}$, while the worst performance occurred in the search for the target in $IR2-I_{5}$, in which the average accuracy rate obtained was only 47%. Figure 15c shows the performance of the system for the configuration with the use of the coprocessor operating in pipeline mode. There was an improvement in response time compared to the use of the coprocessor operating in serial mode. The best performances occurred for the search for the targets in images $IR3-I_{6}$ and $Truck-I_{7}$, producing the lowest average times and highest success rates. The worst performance occurred during the system execution for image $IR2-I_{5}$, resulting in an average success rate of 49.6%.

Figure 16 regards the performance of the usage of the FFA technique. Figure 16a presents the results of the system without the use of a coprocessor. The processing times obtained were above the restriction of 33 ms, which indicates the need for system improvement.

The success rate was above 79% in all image target tracking, and the best system performance occurred for images $IR1-I_{4}$, $IR3-I_{6}$, and $Truck-I_{7}$, resulting in lowest processing times, lowest numbers of iterations, and highest success rates. On the other hand, the worst average search time occurred for the image $IR2-I_{5}$, which yielded a success rate similar to those of the other images due to the complexity of the objective function for this image.

Figure 16b displays the results of the system with the use of a serial architecture coprocessor. There was an improvement in processing time, and the processing time goal was achieved for seven images; it was only not reached for image $IR2-I_{5}$ (34.24 ms). The target was found more quickly in $IR1-I_{4}$; an average of 24.76 ms. All average hit rates were higher than 80%. Figure 16c exhibits the performance of the system in the configuration with the use of the coprocessor operating in pipeline mode. Regarding the average time, there was an improvement compared to the serial mode, reaching the target again for seven images. Only image $IR2-I_{5}$ remained above the target, at 33.8 ms. However, the average success rate was above 80%, meeting the acceptance tracking objective. The best performances occurred for images $IR1-I_{4}$ and $IR3-I_{6}$, resulting in the lowest average processing times. The highest average success rates occurred for images $IR3-I_{6}$ and $Sedan-I_{3}$.

Figure 17 regards the performance of the usage of the FWA technique. Figure 17a presents the results of the system without the use of a coprocessor.

The obtained average times were above the goal of 33 ms, so it is necessary to improve the system. The average success rates were higher than 60%, except for image $IR2-I_{5}$, which achieved only 57.34%.The best performances refer to the searches in images $IR3-I_{6}$, $IR1-I_{4}$, and $Rcar-I_{8}$, which achieved average success rates of at least 90%. Figure 17b shows the results of the system with the use of the coprocessor operating in serial mode. There was an improvement in the average processing time, but the values were still higher than 33 ms. The average success rate achieved met the goal of 60%, except, once again, for image $IR2-I_{5}$, which reached only 57.66%. Figure 17c displays the performance of the system when using the configuration with the coprocessor operating in pipeline mode. The results were close to those obtained with the coprocessor operating in serial mode. The best average time performance occurred for image $IR1-I_{4}$; 50.75 ms, above the 33 ms target. The best average accuracy rate was 97.02%, for image $IR3-I_{6}$. The system performance was the worst for the image $IR2-I_{5}$, for which the search did not achieve the target values for either time or accuracy rate, resulting in mean values of 95.21 ms and 57.02%, respectively.

Figure 18 regards the performance of the usage of the PSO technique. Figure 18a presents the results of the system without the use of a coprocessor.

The obtained average times were above the goal of 33 ms, thus calling for system improvement to reach the required timing restriction. The average success rates were higher than 60% for all considered images. The best performances refer to the searches in images $Sedan-I_{3}$, $IR1-I_{4}$, $IR3-I_{6}$, $Truck-I_{7}$, and $Rcar-I_{8}$, achieving average success rates of at least 99%. It is noteworthy to mention that an acceptance rate of 100% was reached for image $Truck-I_{7}$. Figure 18b shows the results of the system with the use of the coprocessor operating in serial mode. There was a huge improvement in the average processing time. However, some of these values were still higher than 33 ms. Nonetheless, the time values were close to the imposed threshold, which was met for images $Sedan-I_{3}$ and $IR1-I_{4}$. The average success rate achieved met the goal of 60% for all considered images, which reached at least 94%. The best performance was yielded for images $IR1-I_{4}$ and $Truck-I_{7}$, which reached 99.70%. Figure 18c shows the performance of the system when using the configuration with the coprocessor operating in pipeline mode. All the results met the timing restriction of 33 ms. The requirements in time processing were almost half those obtained by the serial configuration system. The best performance in average time occurred for image $Sedan-I_{3}$; 14.26 ms. Moreover, all acceptance rates were above the imposed threshold of 60%, but one can note that, overall, the hit rates were lower than those obtained for the serial configuration. The best average accuracy rate was 99.2% for image $Sedan-I_{3}$. The worst system performance regarding acceptance rate was 89.4%, which image $IR2-I_{5}$ yielded.

### 6.5. Performance Comparison

Figure 19 regards the average performance comparison of the proposed tracking system considering the three investigated configurations. Figure 19a presents the results of the system regarding the processing time.

For the software-only configuration, the CS and ABC techniques required less time to find the target, even compared to PSO. The EHO and FWA techniques took more time compared to the others. For the serial co-design configuration, the results show a reduction in the average processing time to find the target. Only the CS and ABC techniques achieved the goal of 33 ms for all reference images. The FFA technique achieved it for seven out of the eight images considered. For the parallel co-design configuration, the CS, ABC, and PSO techniques achieved the goal of an average time of 33 ms for all reference images. Again, the FFA technique achieved the goal of an average time for only seven out of the eight images considered. The other techniques did not achieve times lower than the goal, and the worst performances were obtained by FWA and EHO.

Figure 19b presents the results of the system regarding the required iterations. For the software-only configuration, the techniques required approximately the same number of iterations to find the target, except for FFA, which required much fewer iterations. For the serial co-design configuration, the results show that the techniques again employed a similar number of iterations to find the target, except for the FFA technique, which required, on average, much fewer iterations to find the object of interest. For the parallel co-design configuration, the FFA technique again required fewer iterations to find the target than the others, which required a similar number of iterations.

Figure 19b presents the results of the system regarding the required iterations. For the software-only configuration, all three system configurations with search techniques achieved the goal for the images. The CS and ABC techniques provided the highest average success rates, although they were lower than those obtained by the PSO technique for the same stopping criteria. For the serial co-design configuration, the results show that all implementations of the search techniques achieved the target goal of 60%. The CS and ABC techniques again provided the highest average success rates, but they were lower than those obtained by the PSO technique. For the parallel co-design configuration, the CS technique achieved the target goal of 60% for all reference images, and the success rates exceeded 90% for two of the eight images considered. The ABC search technique also achieved the target goal for all images, and the average success rates exceeded 85% for six of them. The EHO technique exceeded the 60% target for finding the targets in all reference images. The BFOA technique only achieved the minimum success rate target for seven images, for the chosen parameter configuration, falling below 60% in the search for one of the eight tested images. The FFA technique provided results above 80% for all images, achieving results superior to 91% for two of the images. The FWA technique exceeded the target for seven images. The success rate obtained by the PSO technique was more advantageous for the three investigated configurations.

## 7. Further Discussion

The techniques that showed an accuracy below 60% could still be useful with the parameter settings employed for target localization, as long as another tool is used in conjunction to reinforce the decision-making process. The system used with this technique would need to process an image of the environment with a predefined target to obtain the accuracy rate that would be used as a weight for decision making. The weight would be proportional to the average rate. In this way, a cross-validation process can be applied. To achieve higher accuracy rates, the parameters need to be altered based on the image in which the worst performance is observed, for use with a coprocessor operating in parallel mode. However, these configuration changes may result in worse processing times.

In order to demonstrate the effect of each technique’s average processing time when applied to applications that require a higher degree of certainty, i.e., closer to 100%, the techniques underwent an empirical parameter reconfiguration to attain an average accuracy rate of at least 90% for all eight defined reference images. Table 1 lists the parameters of each technique, whether their values were altered or maintained. It is noteworthy that all the parameter settings of the PSO technique were kept unchanged as the achieved acceptance rates for all images were already above 90%.

Consequently, new average values of processing time and iterations were obtained. It is clear that increasing the number of swarm individuals results in more objective function evaluations per iteration cycle, leading to longer processing times. However, it also tends to position individuals closer to the global optimum, increasing the average accuracy rate. It should be noted that the coprocessor configuration operating in pipeline mode is limited to a maximum of 50 agents given the area resources available on the used board. Therefore, a new performance analysis was conducted regarding the software-only and the serial co-design configurations as the required number of agents was above the allowed 50, except for ABC and FFA. Therefore, for all six techniques, we analyzed the achieved performance based on the processing time vs. the acceptance rate. Nonetheless, for ABC and FFA, it was possible to also analyze the performance of the parallel co-design configuration as the number of agents in these two cases was smaller than 50, i.e., 21 and 17, respectively.

The results for the average processing time obtained without using the coprocessor for 5000 repetitions are displayed in Figure 20a. In every case, the obtained times significantly exceeded the established target of 33 ms, rendering it impractical for use in applications requiring 30-frames-per-second videos. However, all the acceptance rates were either close to or exceeded 90%, as shown in Figure 20b.

The results for the average processing time obtained using the serial architecture coprocessor are shown in Figure 20c. The times were found to be smaller than those obtained with the previous parameter configuration for the techniques. Among all the techniques, only the CS technique achieved processing times below the target of 33 ms for all images. The remaining techniques exceeded the target, and the FWA technique showed the worst performance. The FFA technique provided processing times below 40 ms for six images. However, considering the results for the average accuracy rate obtained using the coprocessor operating in serial mode, which are shown in Figure 20d, all the obtained average rates were above 90%, with the exception of the FFA technique, which showed a rate of 89.94% for image $IR2-I_{5}$ with a standard deviation of 3.29%. The image that provided the highest accuracy rates for target detection is $IR3-I_{6}$, while $IR2-I_{5}$ yielded the lowest average rates.

As mentioned before, the new parameter settings for ABC and FFA allow for its use with the coprocessor operating in pipeline, as the number of agents was less than 50, which is a hardware restriction. Figure 21 exhibits the results regarding time and hit rate.

For ABC, the accuracy rates were all above 90%, but the target average time of 33 ms was not achieved. The worst average performance for target detection occurred for the images $Pickup-I_{2}$, $IR2-I_{5}$, and $Cars-I_{1}$. For FFA, all accuracy rates were also above 90%. However, the average time was above 33 ms for all images. The worst average performance in finding the target occurred for the images $Pickup-I_{2}$, $IR2-I_{5}$, and $Cars-I_{1}$. For image $IR2-I_{5}$, the obtained average accuracy rate was 88.98% with a standard deviation of 3.04%. ABC and FFA had similar performances, while PSO was better in terms of processing time.

We also compared the detection and tracking performance of the proposed system using a pipelined co-design based on PSO to that obtained by works wherein machine learning strategies were used. Table 2 presents the accuracy, precision, recall, and F1-score for the VIVID (pktest01) and VEDAI datasets [70]. Note that the processing time, which is the main issue in this work, was not taken into consideration for all compared works. It is evident that the proposed detection approach yielded a similar performance to that achieved by the machine-learning-based methodology.

## 8. Conclusions

The focus of this research work regarded detecting target objects in images. The research specifications mandate that the system be integrated into a device and exhibit satisfactory real-time performance in videos with 30 frames per second. Moreover, high accuracy rates are necessary to guarantee response quality, prevent false positives, and enable correct decision making by the system operators.

The identification of a target in an image relies on some unique feature of the target and the identification methodology used. In this work, the template matching technique was adopted to analyze image patches and compare them with the target. The Pearson correlation coefficient was calculated to determine the similarity between the target and the cutouts, leveraging its advantages in case of brightness variations in the image. A hardware design was proposed to compute the correlation coefficient efficiently.

The designed system provides three different equipment configurations. The first configuration exploits the Zynq XC7Z015 general-purpose processor performing the search and PCC calculation in the software, working in conjunction with the implemented search technique. The second configuration involves the Zynq processor performing the search in conjunction with a dedicated coprocessor for PCC calculation, operating in serial mode. The third configuration involves the Zynq processor performing the search and the dedicated coprocessor operating in pipeline mode for PCC calculation.

The study found that utilizing a co-design approach with FPGA and intelligent search techniques is an effective means of performing target detection and tracking in images. Depending on the chosen technique and configuration, the system can be applied to real-time video with 30 frames per second. As expected, the best processing time results were achieved using a coprocessor operating in pipeline mode, validating the approach.

The CS and ABC techniques met the processing time and accuracy rate objectives for all images when used with a coprocessor. Nonetheless, CS provided a superior performance in processing time, while ABC obtained higher average accuracy rates. The FFA technique achieved processing times below 33 ms for seven images, achieving average accuracy rates above 80% and requiring few iterations. However, the EHO technique required processing times above 33 ms, achieving average accuracy rates close to 60% even with the adopted configuration. The BFOA technique offered average accuracy rates over 90% for two images, but its timing performance was above 33 ms for all the images. The FWA also presented processing times above 33 ms, achieving average accuracy rates above 95% for one image but below 60% for another.

The image $IR2-I_{5}$ required the highest processing time and iterations due to its complexity regarding the adopted objective function and the presence of many local optima. Overall, the PSO technique proved to be faster and more accurate than the other search techniques. Nonetheless, of the implemented techniques, the CS method occasioned a faster processing time than PSO but a lower average accuracy rate.

The new parameter configuration revealed that achieving an accuracy rate of at least 90% results in inadequate processing time for videos with 30 frames per second. However, these configurations can be useful in other scenarios where the accuracy rate is critical, such as pattern recognition in medical images or applications in videos with fewer frames per second.

In future work, we intend to apply changes to the template, such as rotations, occlusion, and scale variations. Then, we plan to to improve the coprocessor design to handle more than 50 elements in the swarm and to receive multiple frames instead of just one. This way, it is possible to improve the search performance while being able to update the template for each frame, further facilitating tracking. We also suggest a parallel approach wherein two coprocessors can cooperate: one performs video tracking, while the other calculates the target’s future trajectory or even guides the camera, capturing images from the search environment.

## Figures and Tables

**Figure 1 sensors-23-05881-f001:**
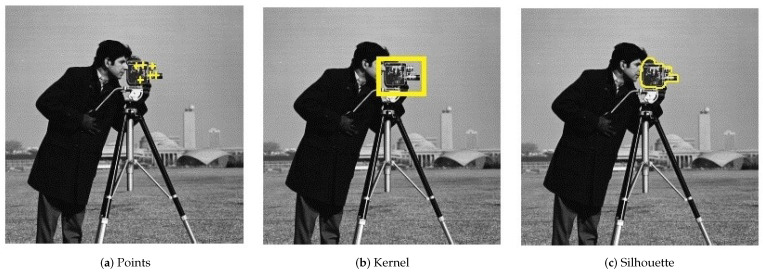
Illustration of existing tracking techniques.

**Figure 2 sensors-23-05881-f002:**
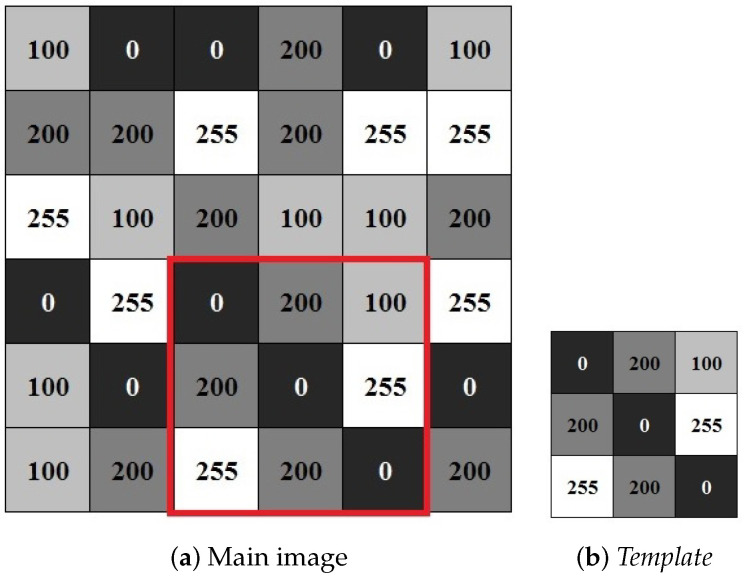
Byte matrices representing the frame and template images in black and white.

**Figure 3 sensors-23-05881-f003:**
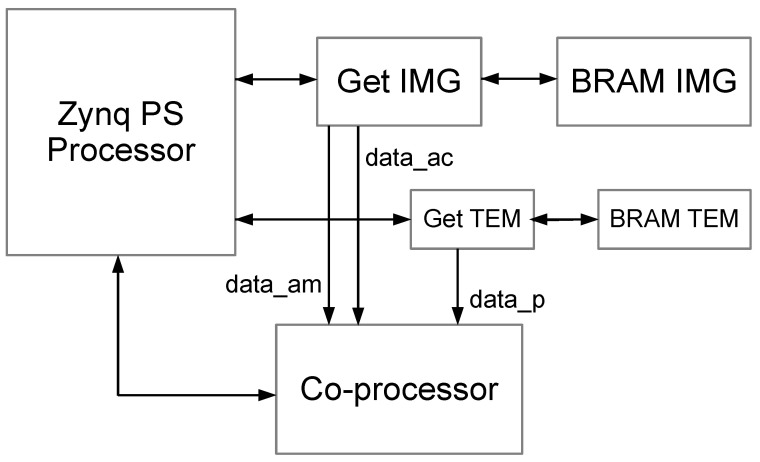
Macro-architecture of the proposed system.

**Figure 4 sensors-23-05881-f004:**
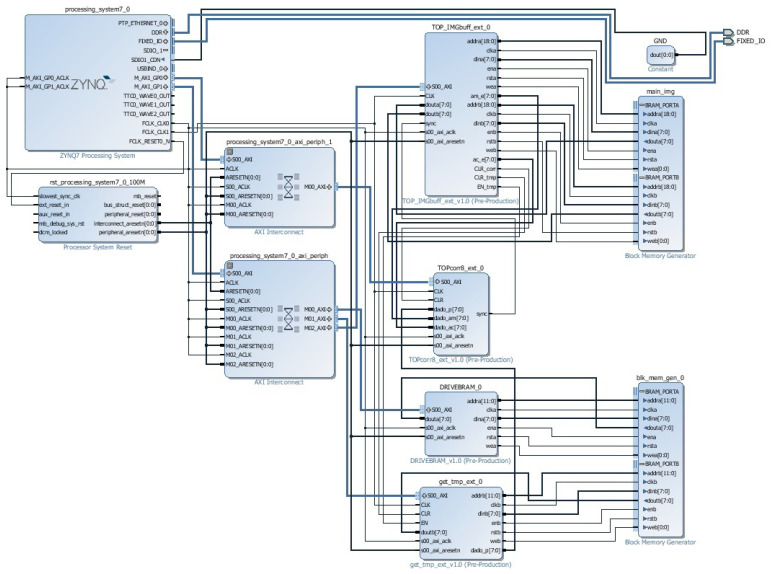
Proposed system when implemented in Vivado.

**Figure 5 sensors-23-05881-f005:**
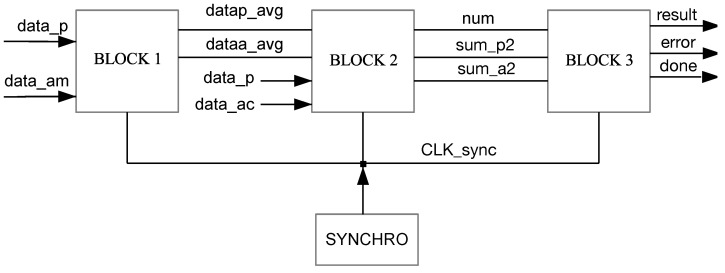
Macro-architecture of the coprocessor.

**Figure 6 sensors-23-05881-f006:**
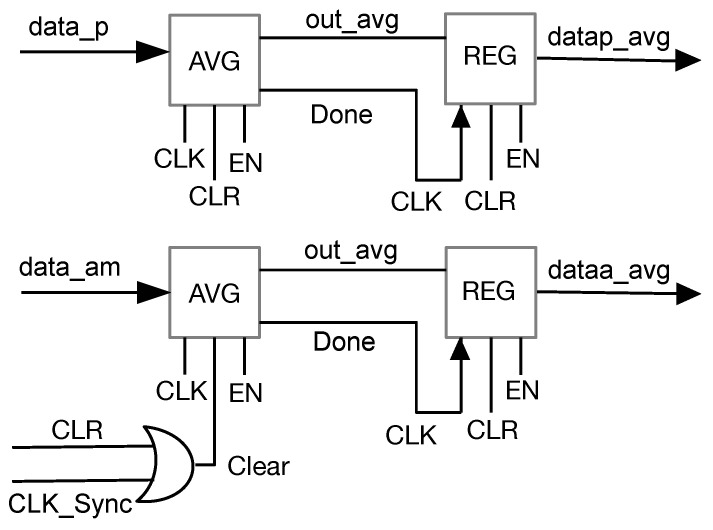
Micro-architecture of component Block 1.

**Figure 7 sensors-23-05881-f007:**
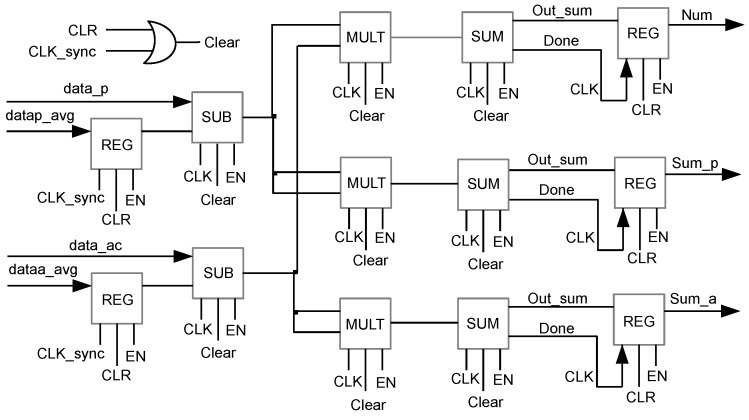
Micro-architecture of component Block 2.

**Figure 8 sensors-23-05881-f008:**
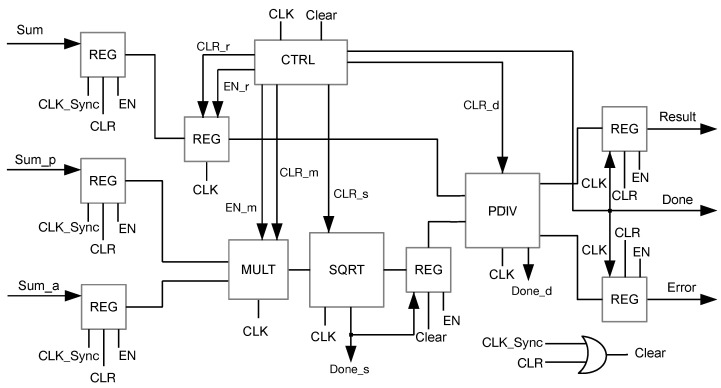
Micro-architecture of component Block 3.

**Figure 9 sensors-23-05881-f009:**
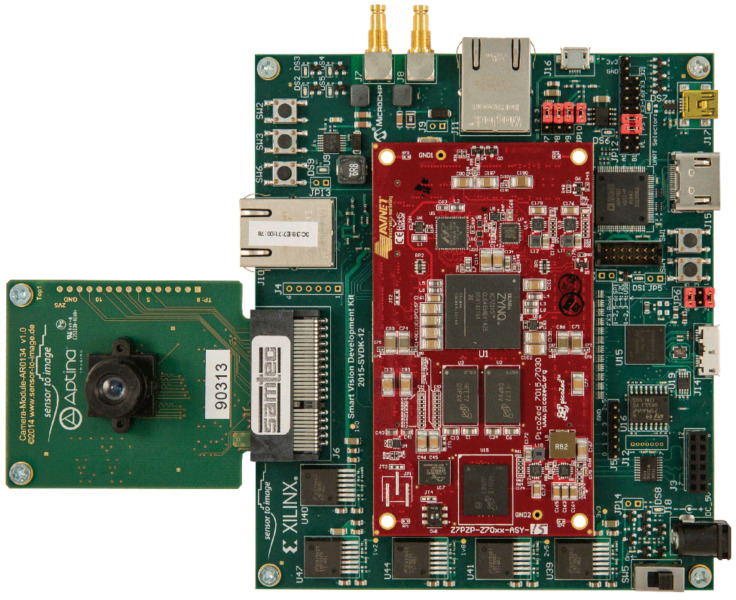
Development board.

**Figure 10 sensors-23-05881-f010:**
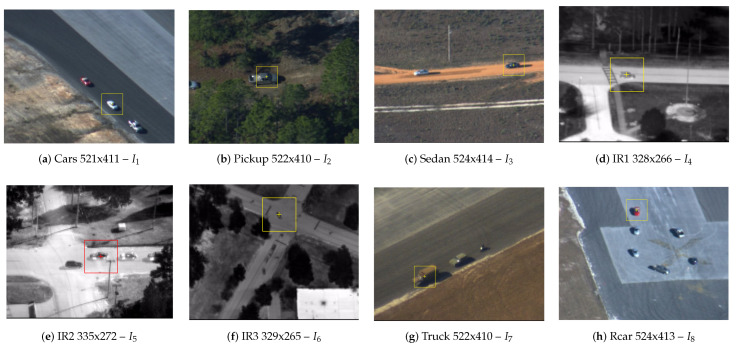
Reference images used as the evaluation dataset.

**Figure 11 sensors-23-05881-f011:**
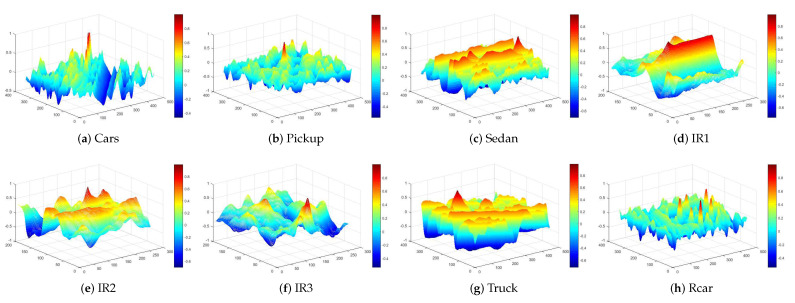
Objective function behaviors within the search space for each reference image.

**Figure 12 sensors-23-05881-f012:**
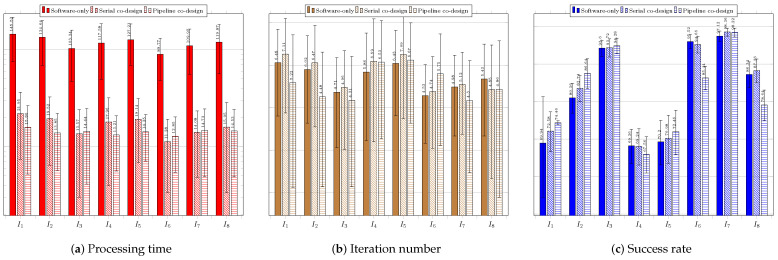
Performance results obtained by CS.

**Figure 13 sensors-23-05881-f013:**
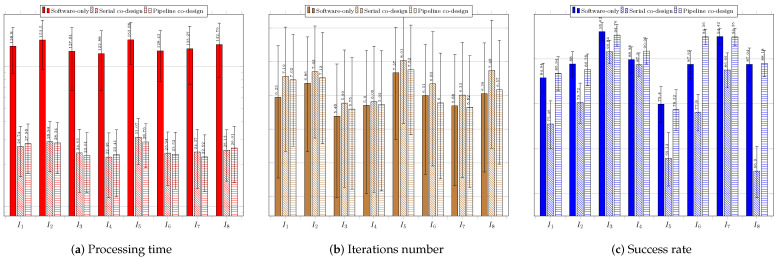
Performance results obtained by ABC.

**Figure 14 sensors-23-05881-f014:**
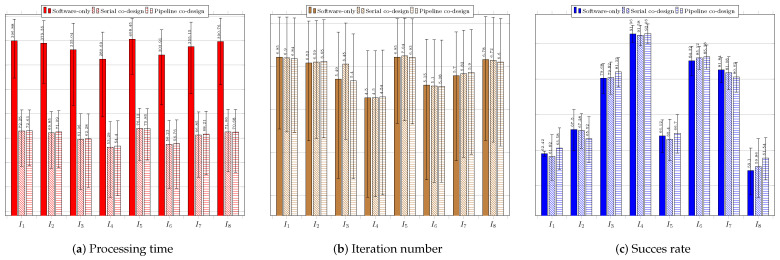
Performance results obtained by EHO.

**Figure 15 sensors-23-05881-f015:**
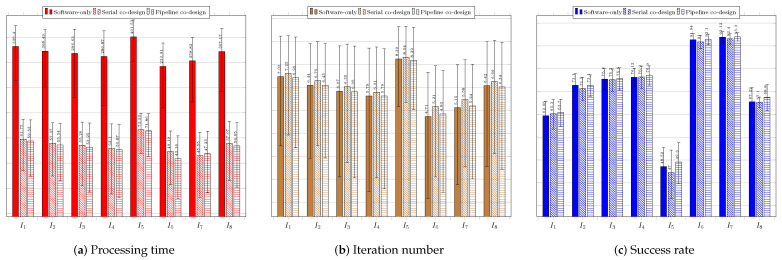
Performance results obtained by BFOA.

**Figure 16 sensors-23-05881-f016:**
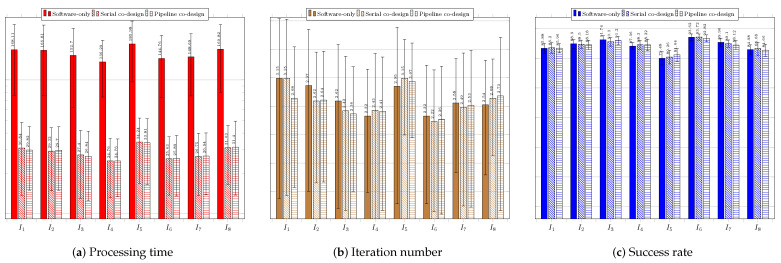
Performance results obtained by FFA.

**Figure 17 sensors-23-05881-f017:**
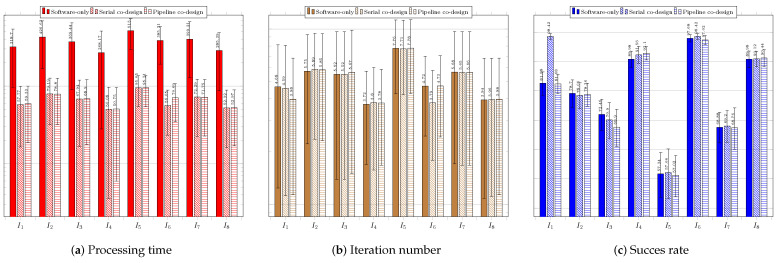
Performance results obtained by FWA.

**Figure 18 sensors-23-05881-f018:**
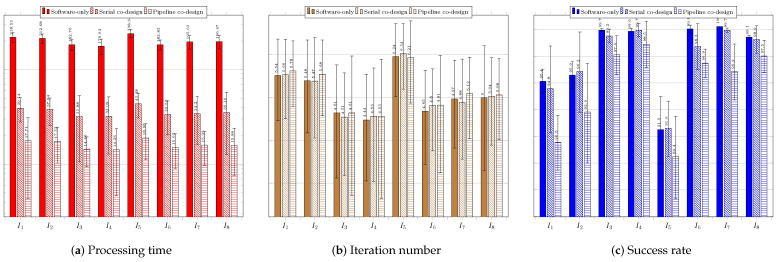
Performance results obtained by PSO.

**Figure 19 sensors-23-05881-f019:**
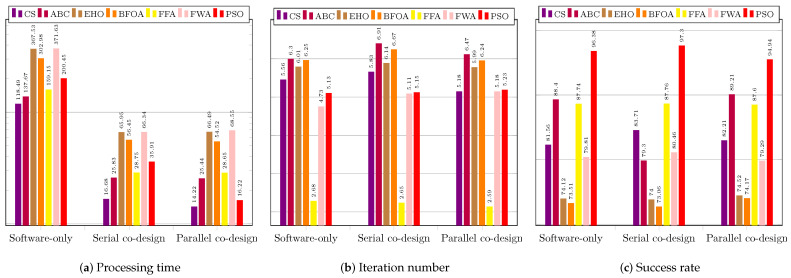
Average performance comparison of the three investigated configurations regarding processing time, iteration, and acceptance rate.

**Figure 20 sensors-23-05881-f020:**
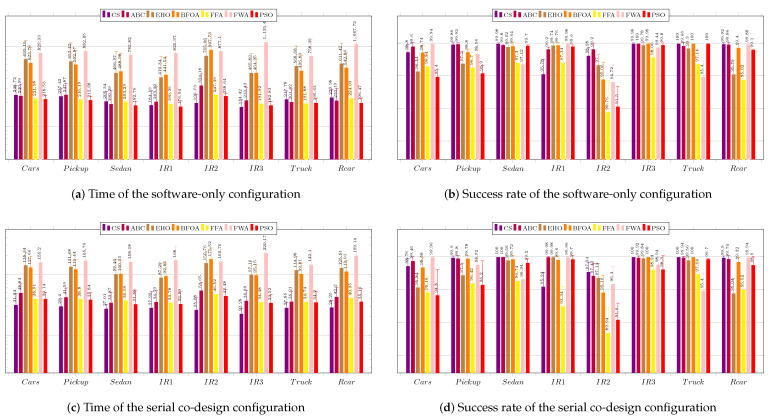
Performance comparison of the software-only and serial co-design configurations for at least a 90% acceptance rate.

**Figure 21 sensors-23-05881-f021:**
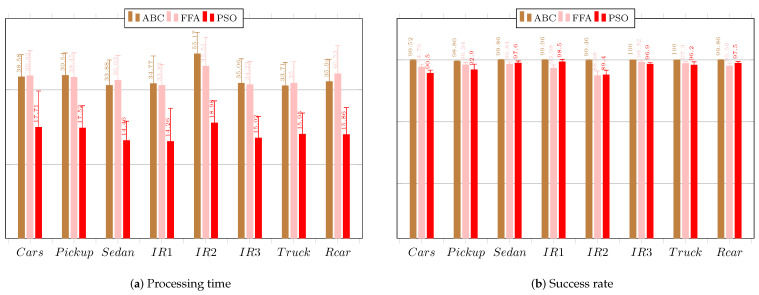
Performance results of the pipeline co-design obtained by ABC and FFA when pursuing at least a 90% acceptance rate.

**Table 1 sensors-23-05881-t001:** New parameter settings for the search techniques.

Technique	Adjusted Parameter	Value	Maintained Parameter	Value
CS	$\;\;\;\;N_{c}\;\;\;\;$	250	$\;\;\;\;P_{a}\;\;\;\;$	25%
			λ	15
ABC	$N_{a}$	21	$N_{esg}$	1
			$E_{exp}$	5
EHO	$N_{e}$	105	α	2.75
	$N_{cla}$	15	β	0.001
BFOA	$N_{b}$	65	$N_{ed}$	1
			$N_{re}$	1
	$P_{ed}$	30%	$N_{chemo}$	1
FFA	$N_{v}$	17	β	1.60
			γ	0.0005
FWA	$N_{f}$	200	*B*	1
	*A*	0.01		
	*K*	800	ξ	0.001

**Table 2 sensors-23-05881-t002:** Performance comparison of proposed system with existing systems wherein machine learning was used for object detection and tracking.

Dataset	Work	Accuracy (%)	Precision (%)	Recall (%)	F1-Score (%)
VIVID	[57]	–	94.30	97.00	96.10
	[55]	95.73	96.77	96.49	97.24
	Proposed work	95.09	96.56	98.13	97.34
VEDAI	[71]	–	89.60	91.50	90.50
	[72]	88.10	79.60	–	89.20
	[55]	92.06	93.19	92.60	93.38
	Proposed work	91.36	90.88	91.31	91.09

## Data Availability

The datasets used in this research are publicly available as indicated in the manuscript.

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
