# Peer review of "Co-Design Dedicated System for Efficient Object Tracking Using Swarm Intelligence-Oriented Search Strategies"

_sensors, 2023, doi:10.3390/s23135881_

Round 1

Reviewer 1 Report

The work presents co-design dedicated system for efficient object tracking using swarm intelligence-oriented search strategies. Concretely, this work proposes to implement template matching in a co-design system and investigating six different swarm intelligence techniques to accelerate the search process. There is a certain of innovation for this paper. However, some problems should be addressed.

1 A discussion section is strongly suggested to add.

2 The contributions are not clear. It is suggested that the authors rewrite this part.

3 In the section of References, the journal abbreviations should be given, rather than the full title of the journal.

4 Some recent progress should be considered in the introduction, such as doi: 10.3390/rs14081784, 10.1109/JSTARS.2022.3229834, and 10.3390/rs12010143.

No

Author Response

Letter to Reviewer #1

Q1: A discussion section is strongly suggested to add.

A1: Further discussion is now highlighted in a separate section. (Please, see Section 7.)

Q2: The contributions are not clear. It is suggested that the authors rewrite this part.

A2: This has now been emphasized. (Please, see he penultimate paragraph of the introduction.)

Q3: In the section of References, the journal abbreviations should be given, rather than the full title of the journal.

A3: I could abbreviate some words, such as Journal (J.) International (Int.). I couldn't find all cited journal abbreviations such that the identification of the journal stays unambiguous.

Q4: Some recent progress should be considered in the introduction, such as doi: 10.3390/rs14081784, 10.1109/JSTARS.2022.3229834, and 10.3390/rs12010143.} 

A4: Done. (Please, see references [2], [3] and [10].)

Last but not least, we would like to take this opportunity to express our gratitude for the valuable comments and the time spent to help us to improve the paper quality!

Reviewer 2 Report

This paper designs a tracking method based on template matching. This paper provides a relatively adequate workload and presents the theoretical basis in detail. However, there remain issues that need to be addressed in this paper:

1.       The authors introduce the speed of the proposed tracker at the end of the abstract. It is suggested to supplement the performance on accuracy and precision of the algorithm in the abstract.

2.       Whether the experimental data in this paper is the data collected by the development board or the publicly available video clips?

3.       It is suggested to add more tracking related metrics such as Robustness, EAO and AR rank in this paper to measure the performance of the model.

4.       Compared with deep learning based trackers such as SiamFC series and SiamRPN series, whether the proposed algorithm can gain an advantage.

Minor editing of English language required

Author Response

Letter to Reviewer #2,

Q1: The authors introduce the speed of the proposed tracker at the end of the abstract. It is suggested to supplement the performance on accuracy and precision of the algorithm in the abstract.

A1: Done. (Please, see the last 3 sentences of the abstract.)

Q2: Whether the experimental data in this paper is the data collected by the development board or the publicly available video clips?

A2: The used dataset is publicly available for testing tracking methodology. The site has now been provide explicitly. (Please, see the 2nd sentence of the 1st paragraph of Section 6.2.)

Q3: It is suggested to add more tracking related metrics such as Robustness, EAO and AR rank in this paper to measure the performance of the model.

A3: The system performance is measured here by the success rate to achieve the main goal that consists of tracking the target object in the incoming video frame at a speed of 30 frames per second (i.e. at most in 33 ms) with an average tracking similarity coefficient of 95%. The latter is actually evaluates how overlapping the patch with the detected object. Here, we propose a hardware that guarantee this kind of metrics. The dataset used is diverse. It contains occluded objects. However, we haven't concentrated on this aspect so that we could really evaluate the robustness of the proposed system. WE intend to do as a future work. We have now mentioned this in the conclusions. (Please, see the last paragraph of Section 8.) Furthermore, the hardware for some of the search techniques was able to guarantee the goal while for other techniques this was not possible. This is discussed in details for each investigated technique. Computing more metrics would stretch further the paper, which is already long. We count on your understanding. (Please, see the details given in Section 6.4.)

Q4: Compared with deep learning based trackers such as SiamFC series and SiamRPN series, whether the proposed algorithm can gain an advantage.

A4: The proposed method does not require any prior training. Moreover, when implemented on hardware, machine learning based tracker require significant hardware area to be implemented. Unlike a possible hardware implementation based on machine learning, the proposed coprocessor is much lightweight in terms of time, power and hardware are requirements. Nonetheless, we compare the accuracy, precision, recall and F1-score obtained by the proposed system against object tracking using neural networks. We show that the proposed system can reach similar performance. (Please, see the last paragraph of Section 7 together with Table 2.)

Last but not least, we would like to take this opportunity to express our gratitude for the valuable comments and the time spent to help us to improve the paper quality!

Reviewer 3 Report

This work proposes an algorithm for for Efficient Object Tracking Using Swarm Intelligence-Oriented Search Strategies. The application is important, but there are several major concerns need to be addressed:

1- The technical novelty of this work is very limited. Ideally a lot more technical novelty is expected.

2- The authors should better explain how the model parameters are tuned.

3- There should be a more detailed ablation study presented in this paper. 

4- The authors should mention the application of this work for some of the popular object detection applications, such as face detection, human detection, etc. This can better promote this work.

5- The experimental comparison is very limited in terms of comparing this work with other prominent works, specially deep learning based models.

6- Many of the recent works on object detection, and also their specific applications are missing in introduction, discussions, and references. The authors should do a more detailed literature overview and add more works. Some of them are suggested below:

- "A PSO and BFO-based learning strategy applied to faster R-CNN for object detection in autonomous driving." IEEE Access 7 (2019): 18840-18859.

- "Going deeper into face detection: A survey." arXiv preprint arXiv:2103.14983 (2021).

7- There are several grammatical errors in this manuscript. Please fix them all.

There are major things which needs be improved on high-level first, to get this paper in a better stage.

This paper needs to improve in several aspects.

Author Response

Letter to Reviewer #3

Q1: The technical novelty of this work is very limited. Ideally a lot more technical novelty is expected.

A1: The novelty of the work is an effective yet efficient real-time hardware system for object tracking. The device can be embedded into any bigger equipment. The contribution of the work has now been better explained. (Please, see the penultimate paragraph of the introduction.)

Q2: The authors should better explain how the model parameters are tuned.

A2: The parameters of the swarming search techniques were configured after many simulations that allowed a systematic search of the parameter space. This is emphasized in the text before the parameters are given. (Please, see the 1st sentence of the 5th paragraph of Section 6.3.)

Q3: There should be a more detailed ablation study presented in this paper. 

A3: This has now been done. (Please, see the 1st paragraph of the introduction and the last three paragraphs of Section 3.)

Q4: The authors should mention the application of this work for some of the popular object detection applications, such as face detection, human detection, etc. This can better promote this work.

A4: Done. (Please, see the first paragraph of the introduction.)

Q5: The experimental comparison is very limited in terms of comparing this work with other prominent works, specially deep learning based models.

A5: A comparison of the proposed system to existing implementations based on machine leaning regarding detection and tracking accuracy has now been introduced. (Please, see the last paragraph of Section 7 together with Table 2.)

Q6: Many of the recent works on object detection, and also their specific applications are missing in introduction, discussions, and references. The authors should do a more detailed literature overview and add more works. Some of them are suggested below: - "A PSO and BFO-based learning strategy applied to faster R-CNN for object detection in autonomous driving." IEEE Access 7 (2019): 18840-18859; - ""Going deeper into face detection: A survey." arXiv preprint arXiv:2103.14983 (2021).

A6: We have now reviewed and commented on recent works where neural networks are used to detected and classify object, including the ones suggested. (Please, see the first paragraph of the introduction and the last 4 paragraphs of Section 3, with the newly cited works [1], [2], [51], [52], [53], [54].)

Q7: There are several grammatical errors in this manuscript. Please fix them all.

A7: The entire text of the paper has now been reviewed with respect to typos and gramatical aspects.

Last but not least, we would like to take this opportunity to express our gratitude for the valuable comments and the time spent to help us to improve the paper quality!

Round 2

Reviewer 1 Report

The authors have addressed my concerns. 

Author Response

Letter to Reviewer #1

Best regards,

Q1: The authors have addressed my concerns.

A1: Thank you!

Last but not least, we would like to take this opportunity to express our gratitude for the valuable comments and the time spent to help us to improve the paper quality!

Reviewer 2 Report

The author has answered and explained all the questions in the revised paper. I have no further questions.

Author Response

Letter to Reviewer #2

Best regards,

Q1: The author has answered and explained all the questions in the revised paper. I have no further questions.

A1: Thank you!

Last but not least, we would like to take this opportunity to express our gratitude for the valuable comments and the time spent to help us to improve the paper quality!

Reviewer 3 Report

The authors have addressed some of my comments and the paper is improved. Please try to add more experimental comparison with other works, and do a final proofread for the final version.

The authors have addressed some of my comments and the paper is improved. Please try to add more experimental comparison with other works, and do a final proofread for the final version.

Author Response

Letter to Reviewer #3

Best regards,

Q1: The authors have addressed some of my comments and the paper is improved.

A1: Thank you!

Q2: Please try to add more experimental comparison with other works.

A2: The provided comparison of the proposed system to existing implementations based regarding detection and tracking accuracy has now been extended. (Please, see the last paragraph of Section 7 together with Table 2.)

Q3: Please try to add more experimental comparison with other works, and do a final proofread for the final version.

A3: The whole text of the manuscript has now been proof-read.

Last but not least, we would like to take this opportunity to express our gratitude for the valuable comments and the time spent to help us to improve the paper quality!
